# Mixture of Cache-Conditional Experts for Efficient Mobile Device Inference

**Andrii Skliar** *†                                                      *andrii@contextual.ai*
*Contextual AI*

**Ties van Rozendaal** *†                                                    *tvr@tivaro.nl*
*Qualcomm AI Research* ‡

**Romain Lepert**                                                  *romain@qti.qualcomm.com*
*Qualcomm AI Research* ‡

**Todor Boinovski**                                                 *todorb@qti.qualcomm.com*
*Qualcomm AI Research* ‡

**Mart van Baalen**                                                   *mart@qti.qualcomm.com*
*Qualcomm AI Research* ‡

**Markus Nagel**                                                   *markusn@qti.qualcomm.com*
*Qualcomm AI Research* ‡

**Paul Whatmough**                                                *pwhatmou@qti.qualcomm.com*
*Qualcomm AI Research* ‡

**Babak Ehteshami Bejnordi**                                        *behtesha@qti.qualcomm.com*
*Qualcomm AI Research* ‡

**Reviewed on OpenReview:** *https://openreview.net/forum?id=ul4W26KEKz*

## Abstract

Mixture of Experts (MoE) LLMs enhance performance by selectively activating specialized subnetworks ("experts") per input. While MoEs offer efficiency benefits through distributed inference in typical high-throughput settings, deploying them on memory-constrained devices remains challenging, particularly for sequential token generation with batch size one. In this work, we optimize MoE for such constrained environments, where only a subset of expert weights fit into DRAM. Through empirical analysis, we show MoEs can tolerate careful deviations in expert selection with minimal predictive performance loss. Inspired by this observation, we propose a novel cache-aware routing strategy that leverages expert reuse during token generation to significantly improve cache locality. Evaluating on language modeling, MMLU, and GSM8K benchmarks, our method reduces cache miss rates by over 50%, with negligible impact on perplexity (0.1%–3%) and downstream task accuracy (<0.1%). Unlike prior methods limited by the optimal oracle cache bound, our approach surpasses this theoretical limit by allowing slight flexibility in expert selection. Finally, we present on-device results demonstrating 2× speedups on mobile hardware, offering a flexible and training-free solution to extend MoE's applicability across real-world applications.

---

*Equal contribution
†Work conducted while employed at Qualcomm.
‡Qualcomm AI Research is an initiative of Qualcomm Technologies, Inc.

# 1 Introduction

Mixture of Experts (MoE) models have emerged as a powerful approach in large language models (LLMs), offering advantages in scalability and efficiency (Fedus et al., 2022b;a). By selectively activating a subset of experts for each input, MoEs handle diverse data distributions and capture complex patterns more effectively than traditional dense models. Recent models such as DeepSeek-V3 (DeepSeek-AI et al., 2024), Qwen2.5-Max (Yang et al., 2024), GPT-4 (OpenAI et al., 2024),

Gemini (Team et al., 2024), and Mixtral (Jiang et al., 2024) provide evidence for MoEs' success in leveraging specialized sub-networks to achieve superior performance. Simultaneously, small language models (SLMs) with MoE architectures, such as OLMoE (Muennighoff et al., 2024), Phi-3.5-MoE (Abdin et al., 2024), and Qwen-MoE (Qwen Team, 2024), have shown promise for server deployment due to their ability to maintain high performance with fewer active parameters per token.

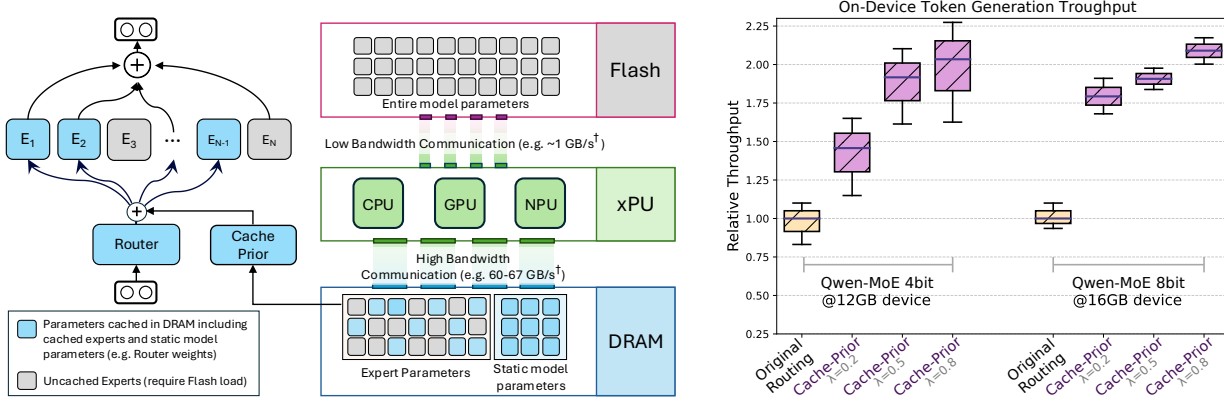

Figure 1: Overview of our proposed cache-aware routing method. (Left) MoE models are hosted in slower Flash storage due to their size, with only a subset of expert weights cached in faster DRAM. Our cache prior method is cache-aware and adjusts expert selection to promote experts already in DRAM, significantly reducing cache misses and improving inference efficiency. (Right) Throughput of our routing method for the 4-bit and 8-bit quantized Qwen1.5-MoE models deployed on two mobile devices with 12GB and 16GB available memory and cache size of 30 and 45 experts, respectively. Our proposed Cache-Aware Routing method significantly enhances the token generation throughput compared to the baseline leveraging a Least Recently Used (LRU) Cache. † `https://en.wikipedia.org/wiki/Universal_Flash_Storage`

Despite their advantages, deploying MoE models on memory-constrained devices like smartphones and laptops presents significant challenges. One primary issue is that the parallelism techniques enhancing efficiency in server deployments of MoEs are less applicable in on-device scenarios, where tokens are generated sequentially with a batch size of one. This limitation prevents batching or parallel processing through expert sharding, leading to increased latency and inefficiencies during inference. Additionally, MoE models have a significantly larger memory footprint than dense architectures, complicating deployment on memory-constrained devices.

These models often exceed available DRAM, and loading model weights directly from flash storage is detrimental to latency. To mitigate this, DRAM can be used as a cache for expert weights, but this approach is only effective if the hit rate is high. Ideally, MoE models should have strong cache locality, with a small set of experts being accessed repeatedly over long token sequences. This requires better strategies for expert selection and cache optimization during inference.

Previous work has tried to improve MoE efficiency by pre-fetching experts based on hidden states from earlier layers and using LRU caching to store recently used experts (Eliseev & Mazur, 2023; Yuan et al., 2024a). However, these methods assume that expert selection follows predictable patterns and has a strong locality. Temporal locality, in the context of MoE caching, refers to the principle that recently accessed experts are likely to be accessed again in subsequent token generations. A key challenge, which we analyze empirically in Section 4.6, is that state-of-the-art MoEs often lack strong temporal locality in expert selection, leading to

inefficient Least Recently Used (LRU) caching strategies due to frequent evictions and reloads, which results in a poor cache hit rate.

In this work, we aim to utilize already-trained, high-performing MoE models by incorporating cache locality priors to make them more efficient for memory-limited devices. Our method increases expert reuse during token generation, improving the cache hit rate. This approach is training-free and can be applied directly to off-the-shelf MoE models, enhancing their efficiency for on-device deployment.

To enable efficient inference on memory-constrained devices, we first analyze the temporal locality in expert selection across four SoTA MoEs. We demonstrate their sensitivity to expert dropping and random expert swapping in Section 2.3, which shows the potential for exploring alternative routing strategies. Motivated by these observations, we propose a method that manipulates router logits to increase the probability of selecting cached experts, all without the need for additional training. In summary, this work makes the following contributions:

- We demonstrate that state-of-the-art MoE models lack temporal consistency in expert selection, which leads to suboptimal latency performance when caching strategies are employed. However, our findings indicate that these models exhibit low sensitivity to variations in the selection of lower-weighted experts, suggesting that lossy routing strategies can be implemented without substantial degradation in overall performance.

- We propose a method to significantly improve the cache hit rate of experts, enhancing throughput for token generation in existing MoEs without requiring additional training. Our approach strikes a balance between cache hit rate (and token generation latency) and model accuracy under a fixed memory budget.

- We report the performance of the proposed routing strategy for language modeling and present evaluation results on MMLU and GSM8K benchmarks. Surprisingly, we show that imposing some degree of routing consistency enhances benchmark scores while making these models more on-device friendly.

- We present on-device results of our proposed approach on two different mobile devices with various memory constraints, demonstrating a consistent speed-up of up to $2\times$ compared to the original routing with LRU caching. Our training-free method is adaptable and can be deployed across a variety of real-world scenarios and diverse hardware configurations, making it suitable for devices with different memory limitations.

## 2 Background and Motivation

In this section, we give an overview of MoE models, followed by a sensitivity analysis that looks at how expert selection affects model performance, showing the potential for alternative routing strategies explored in the next section.

### 2.1 Preliminaries of MoE

The sparse MoE layer consists of $N$ expert networks $E_1, E_2, \ldots, E_N$ and a routing network $G$ that selects a subset of experts based on their relevance to the input token $\mathbf{x} \in \mathbb{R}^d$. The output of the MoE layer during inference is given by:

$$\mathbf{y} = \sum_{i \in \mathbf{r}[:K]} \mathbf{w}_i E_i(\mathbf{x}), \quad (1) \qquad \mathbf{r} = \text{argsort } \mathbf{w}, \quad (2) \qquad \mathbf{w} = \sigma(\mathbf{z}) = \sigma(G(\mathbf{x})). \quad (3)$$

Here, $\mathbf{z} = G(\mathbf{x})$ represents the logits assigned by the router to the experts, $\sigma$ is the softmax function, and $\mathbf{r}$ is a ranking vector of experts based on the weights, from which only the top-$K$ experts are selected. The weights assigned to each expert may be re-normalized after the top-$K$ selection in Equation 1.

In conventional MoEs, the top-$K$ routing mechanism for selecting experts is dynamic and input-dependent, which can cause significant variability in the experts chosen for each token. This inconsistency makes it difficult to achieve high cache hit rates, as it reduces the chance that selected experts are already in the cache. As shown in Table 2, cache lifetimes are short, and cache miss rates are high for all models using a simple LRU caching policy.

## 2.2 MoE Deployment on Memory-Constrained Devices

Deploying MoEs on memory-constrained devices, like mobile phones, is challenging due to limited memory resources. As shown in Figure 1 (Left), these devices typically feature a combination of DRAM and flash storage, each with different speed and capacity characteristics. DRAM offers high bandwidth but limited capacity, while flash storage provides larger capacity at the cost of slower access speeds.

MoE models are typically larger than dense models, therefore, not all parameters may fit into the limited DRAM available on many devices. Static parameters, such as attention weights that do not change during inference, can be stored permanently in DRAM. In contrast, due to the dynamic nature of expert selection, only a subset of experts is loaded into DRAM at a time. To improve efficiency, recently used experts can be retained in a cache, reducing the need to reload them from slower storage. For an effective caching strategy, a high cache hit rate is important because it means experts are quickly available from DRAM, reducing the overall latency. The cache hit rate is defined as the proportion of selected experts that are already in the cache at the time of their selection. Let the cache set $C$ denote the indices of experts currently stored in DRAM, and $\mathbf{r}[:K]$ be the experts selected by the routing network. The cache hit rate can be expressed as:

$$hit\_rate = \frac{\text{count}(\mathbf{r}[:K] \cap C)}{K}. \tag{4}$$

The *miss_rate* is simply $1 - hit\_rate$.

The goal of this work is to improve the expert cache hit rate without affecting the model's predictive performance. To achieve this, we propose functions that modify the ranking vector $\mathbf{r}$ used to select the top-$K$ experts without making any other changes to the MoE forward pass as shown in Equation 1. Our objective is to create new ranking vectors that prioritize experts already in cache while minimizing any negative impact on model performance. This allows for a model-agnostic method that can be universally applied.

## 2.3 MoEs Are Less Sensitive to Dropping Experts with Lower Scores

A key opportunity for making MoE models more cache-friendly without compromising task performance is understanding how sensitive the models are to changes in expert selection. To explore this, we analyze four different MoE architectures: DeepSeek-V2-Lite (DeepSeek-AI et al., 2024), Qwen1.5-MoE-A2.7B (Qwen Team, 2024), Phi-3.5-MoE (Abdin et al., 2024), and Mixtral-8x7B (Jiang et al., 2024). Our results show that these models can tolerate deviations in the expert selection, as long as the experts with the highest weights are retained.

The left plot in Figure 2 illustrates the impact of completely removing (pruning) experts ranked at or above a specified index on Wikitext perplexity. Removing the second highest ranked expert leads to a performance drop across all four models. However, models with more active experts, like Qwen and DeepSeek, recover quickly, allowing higher-ranked experts to be pruned with minimal performance loss.

To examine how the rank of an expert impacts performance while keeping the total number of active experts constant, we replaced the expert ranked $k$ with a randomly selected expert, as shown in the right plot of Figure 2. The results show that swapping the top-ranked expert severely compromises model performance. Replacing the second-ranked expert also causes noticeable degradation, but the model remains functional. Beyond this, in granular MoEs (with a larger number of smaller active experts), the model becomes highly resilient to expert selection changes. This suggests that while top-1 is critical for all MoEs, the top-2 expert in standard MoEs such as Mixtral and Phi-MoE can be swapped without a major loss in perplexity. Granular models maintain flexibility in expert selection and incur minimal performance loss by swapping the 3rd highest ranked expert and beyond.

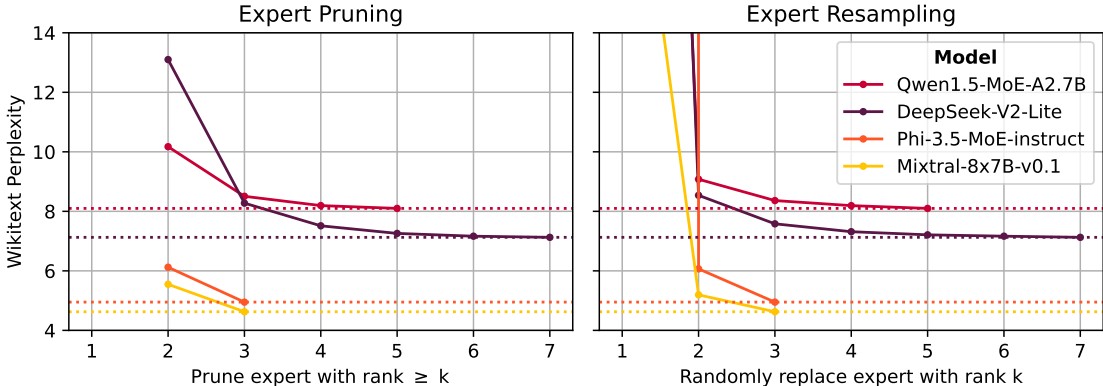

Figure 2: Expert sensitivity analysis. We show the effect of dropping or replacing experts as selected by the router. The x-axis represents the expert rank (ordered by their scores) and the y-axis shows the Wikitext validation perplexity (lower values indicate better performance). The dashed lines represent the baseline perplexity of the MoE models.
The left figure illustrates the effect of dropping all experts ranked higher than $k$, whereas the right figure depicts the impact of randomly replacing the expert at rank $k$.

Interestingly, further analysis (Appendix A) using a greedy search to find the optimal expert combination suggests that the router's predictions are often suboptimal. For Mixtral-8x7B, the router's predicted top-2 experts yield the best performance only 28% of the time on average with a maximum of 38% in the last layer. This supports the idea that lower-ranked experts can be replaced with little impact on performance. Taken together, these findings highlight that while preserving the highest-ranked experts is crucial, MoE models offer flexibility in expert selection—an insight we leverage throughout this work.

## 3 Cache-aware Expert Routing

In this section, we introduce our main method along with two simple baselines: Max Rank and Cumulative Probability Threshold. These methods, progressively more refined, balance cache hit rate with downstream task performance, improving model throughput and efficiency. All of our methods are general and training-free, making them easily applicable to off-the-shelf MoE models for on-device deployment.

### 3.1 Max Rank Approach

In Section 2.3, we demonstrated that MoEs exhibit some robustness to the swapping of their experts with random ones. Building on this insight, we aim to deviate from the standard top-$K$ set of experts $\mathbf{r}[:K]$, selected by the router and instead promote experts that are already present in the cache $C$. We first define a general promotion operation that elevates the ranking of all experts in an ordered subset $\mathbf{r}^{\text{subset}}$:

$$\text{promote}(\mathbf{r}^{\text{subset}}; \mathbf{r}^{\text{all}}) := \mathbf{r}^{\text{subset}} \oplus (\mathbf{r}^{\text{all}} \setminus \mathbf{r}^{\text{subset}}). \tag{5}$$

Here $\oplus$ denotes the concatenation operation and the set subtraction $\setminus$ preserves the order of it's left operand. Importantly, all sets in this context are ordered sets and maintain the order of elements as in their original ranking. This ensures that the promotion operation is well-defined, as it preserves the relative ordering of experts throughout.

A naive solution would be to simply promote all experts currently in the cache, but this approach does not take into account the expert probability assigned by the router. Instead, we limit the promotion of elements in the cache by a maximum allowed rank $M$ such that the experts with the lowest router probabilities are not chosen:

$$\mathbf{r}^{\text{max-rank}} := \text{promote}(\mathbf{r}[: M] \cap C; \mathbf{r}). \tag{6}$$

As observed in Section 2.3, certain top-$J$ experts are crucial for model performance. To ensure these experts are not overwritten by others in the cache, we always select them, even if they are not in the cache. This can be achieved by adding a second promotion operation:

$$\text{promote}\Big(\mathbf{r}[: J]; \text{promote}(\mathbf{r}[: M] \cap C; \mathbf{r})\Big). \tag{7}$$

Finally, the pseudocode for the max-rank algorithm with always keeping the top-$J$ experts is shown in Algorithm 1. Additionally, we provide an intuitive explanation of the algorithm with an example in Appendix B.

---
**Algorithm 1** Max Rank
---
**Require:** max-rank $M$, minimal-rank $J$, and original ranking $\mathbf{r}$
1: $\mathbf{r}' \leftarrow \text{promote}(\mathbf{r}[: M] \cap C; \mathbf{r})$
2: $\mathbf{r}' \leftarrow \text{promote}(\mathbf{r}[: J]; \mathbf{r}')$
3: **return** $\mathbf{r}'$
---

### 3.2 Cumulative Probability Threshold Approach

A limitation of the max-rank routing strategy is that it does not take into account the distribution of all expert probabilities $G(\mathbf{x})$. For instance, an input where the most probable expert has a very high probability will likely require a very low max-rank $M$ to maintain good model performance. Conversely, for an input with uniformly distributed router probabilities, there is no strong expert preference, allowing for a very high max-rank $M$ for better cache hit rates.

We address this issue in the cumulative probability threshold approach by dynamically choosing the max-rank $M$ for every layer and every input $\mathbf{x}$. The process is conceptually similar to the sampling approach by Holtzman et al. (2020) in which tokens are sampled from a dynamic set containing the vast majority of the probability mass. We determine $M$ by summing the sorted probabilities of the router outputs $G(\mathbf{x})$ from highest to lowest until a predefined cumulative probability threshold $p$ is reached:

$$M = \min i \quad s.t. \sum_{j=1}^{i} G(\mathbf{x})[r_j] \geq p \tag{8}$$

Using the set of $M$ values found for each layer and each input, we apply the max-rank strategy from Equation 7 to select a new set of experts. We provide an schematic explanation of this approach in Appendix B. The pseudocode for the cumulative probability threshold approach can be found in Algorithm 2. This method is more dynamic as it accounts for the confidence in the router's predictions. When the set of experts required to reach the threshold is larger, it indicates less confidence from the router (flat distribution), allowing for more flexibility in replacing a non-cached expert with the highest probability expert available within this set that is already in the cache.

### 3.3 Cache-Prior Reranking

Although the cumulative threshold routing strategy dynamically adjusts the max-rank, it still imposes a hard limit beyond which experts are not promoted. This may be suboptimal, especially for router distributions with long tails, which may be better suited for making cache-friendly decisions.

To address this, we use a Cache-Prior that directly manipulates the router logits $\mathbf{z} = G(\mathbf{x})$ to increase the probability of selecting experts already present in cache. Importantly, we use the manipulated logits $\mathbf{z}'$ only

---

**Algorithm 2** Cumsum Threshold

---

**Require:** probability threshold $p$, minimal-rank $J$, original ranking $\mathbf{r}$, and expert weights $\mathbf{w}$

1: $p_{\text{cum}} \leftarrow 0$
2: $M \leftarrow 0$
3: **while** $p_{\text{cum}} < p$ **do**
4:     $M \leftarrow M + 1$
5:     $p_{\text{cum}} \leftarrow p_{\text{cum}} + \mathbf{w}[r[M]]$
6: **end while**
7: $\mathbf{r}' \leftarrow \text{promote}(\mathbf{r}[: M] \cap C; \mathbf{r})$
8: $\mathbf{r}' \leftarrow \text{promote}(\mathbf{r}[: J]; \mathbf{r}')$
9: **return** $\mathbf{r}'$

---

| Model | Params | | Experts | | | | Footprint (int4) | | |
|---|---|---|---|---|---|---|---|---|---|
| | **Active** | **Total** | **Shared** | **Total** | **Top-k** | **Params** | **min** | | **max** |
| Mixtral-8x7B | 13B | 46.7B | | 8 | 2 | 176M | 6,5 GB | - | 23,4 GB |
| Phi-3.5-MoE | 6.6B | 41.9B | | 16 | 2 | 79M | 3,3 GB | - | 21,0 GB |
| DeepSeek-V2 | 2.8B | 15.9B | 2 | (64+2) | (6+2) | 8.6M | 1,4 GB | - | 8,0 GB |
| Qwen1.5-MoE | 2.7B | 14.3B | 4 | (60+4) | (4+4) | 8.6M | 1,4 GB | - | 7,2 GB |

Table 1: The MoE architectures used in our experiments. Column "Experts" / "Params" indicates the number of parameters per expert. Column "Footprint" represents the total size of all model parameters and the expert cache under int4 quantization, with the cache size bounded by $k$ (minimum) and $N$ (maximum).

to find a new ranking vector $\mathbf{r}'$ as defined in Equation 2. We still use the unmodified router logits to compute the expert weights used in Equation 1.

Figure 3 provides an overview of our proposed cache-aware routing method. Let the bitmask $\mathbf{m}_t \in \{0,1\}^N$ represent the state of the cache following the generation of the $(t-1)$-th token, where each bit indicates whether an expert is in the cache or not.

To ensure the top-$J$ experts are always selected, regardless of their presence in the cache, we can optionally add them to the bitmask $\mathbf{m}_t$. This results in an updated bitmask, $\tilde{\mathbf{m}}_t$. Using this updated bitmask, we then boost the logits for the corresponding experts as follows:

$$\mathbf{z}' = \mathbf{z} + \lambda \cdot \Delta_{\text{avg}} \cdot \tilde{\mathbf{m}}_t, \tag{9}$$

where $\lambda \in [0,1]$ is a scaling factor that determines the influence of the cache state on the logits. This scaling factor allows our method to interpolate between the original routing ($\lambda = 0$, equivalent to baseline cache hit rate) and a fully cache-driven selection ($\lambda = 1$, potentially zero cache misses, but higher perplexity). $\Delta_{\text{avg}}$ is defined as:

$$\Delta_{\text{avg}} = \mathbb{E}_{\mathbf{x} \in \mathcal{X}} \mathbb{E}_{t \in 1...T} [\max(\mathbf{z}) - \min(\mathbf{z}))]. \tag{10}$$

which represents the range in logits for this layer that we estimate using a running average over sequences and tokens. In essence, our Cache-Prior method involves adding a fraction of the average logit range to the experts that are already in the cache. By utilizing the true range of weights, our Cache-Prior becomes adaptive, allowing a single hyperparameter to yield distinct effects across different layers and tokens. We provide an ablation on the choice of $\Delta$, including approaches for predicting it directly for each token, in Appendix D and E.

## 4  Experiments

In this section, we evaluate the effectiveness of our Cache-Prior routing method and compare it to out Pruning, Max-Rank, and Cumulative Sum Thresholding baselines, as described in Section 3. We implement all methods on four state-of-the-art MoE models: DeepSeek-V2-Lite (DeepSeek-AI et al., 2024), Qwen1.5-

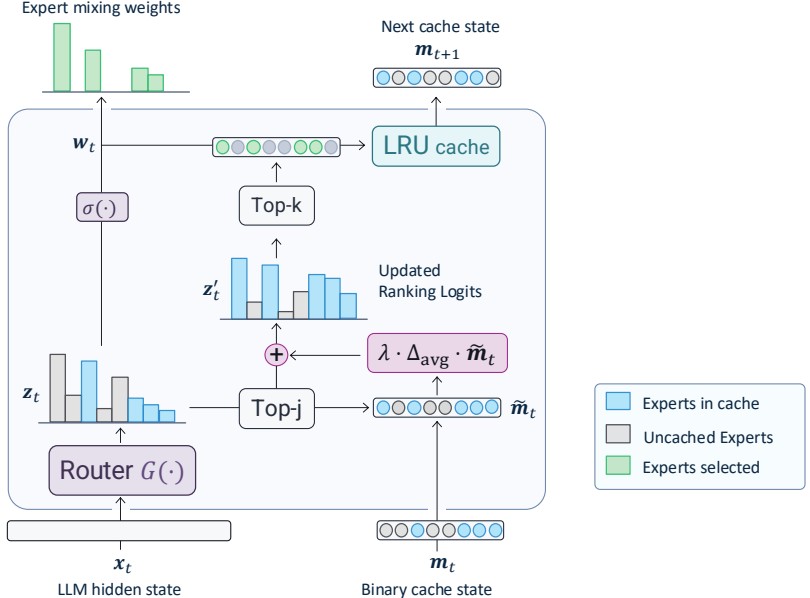

Figure 3: Our proposed Cache-Prior routing method adds a bias to the logits only for in-cache experts $\mathbf{m}_t$, encouraging their selection. The magnitude of the bias is determined by the average logit range, $\Delta_{\mathrm{avg}}$, and the tradeoff parameter $\lambda$. The updated logits, $\mathbf{z}'_t$, are used only for re-ranking experts, while the expert weights, $\mathbf{w}_t$, remain unchanged.

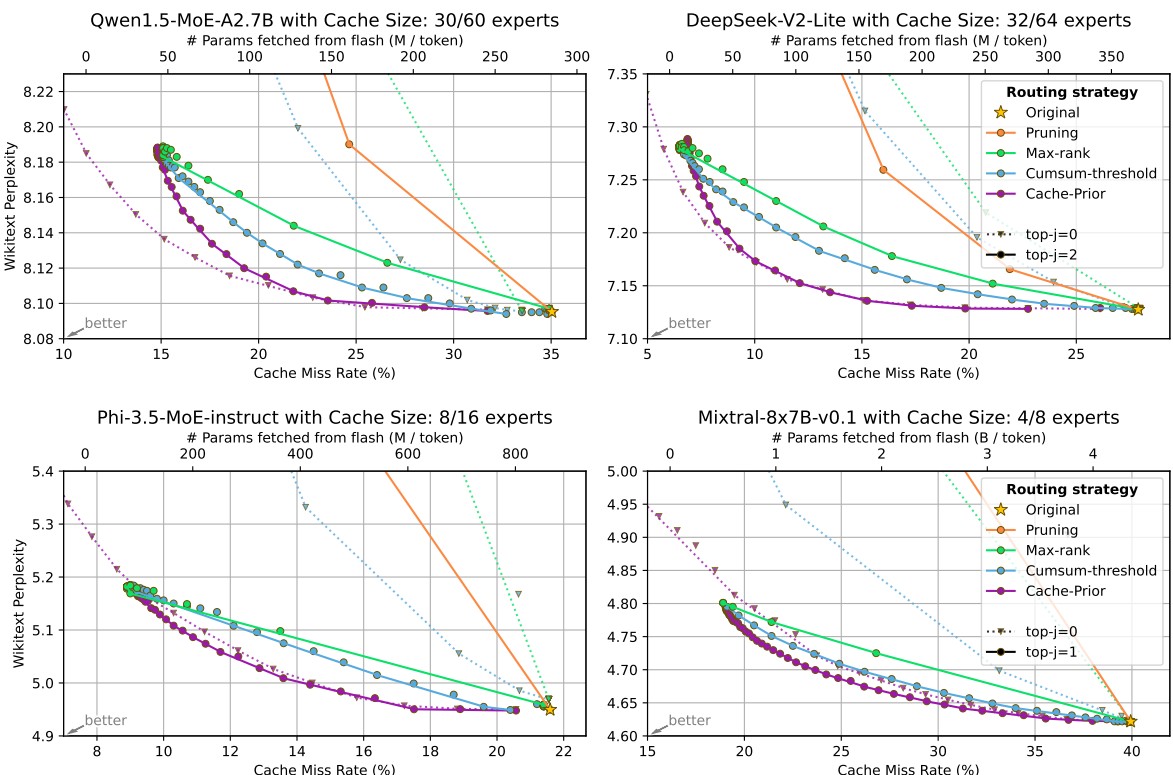

Figure 4: Trade-off curves between Wikitext perplexity and cache miss rate for four MoE models with a cache size set to half the total number of experts.

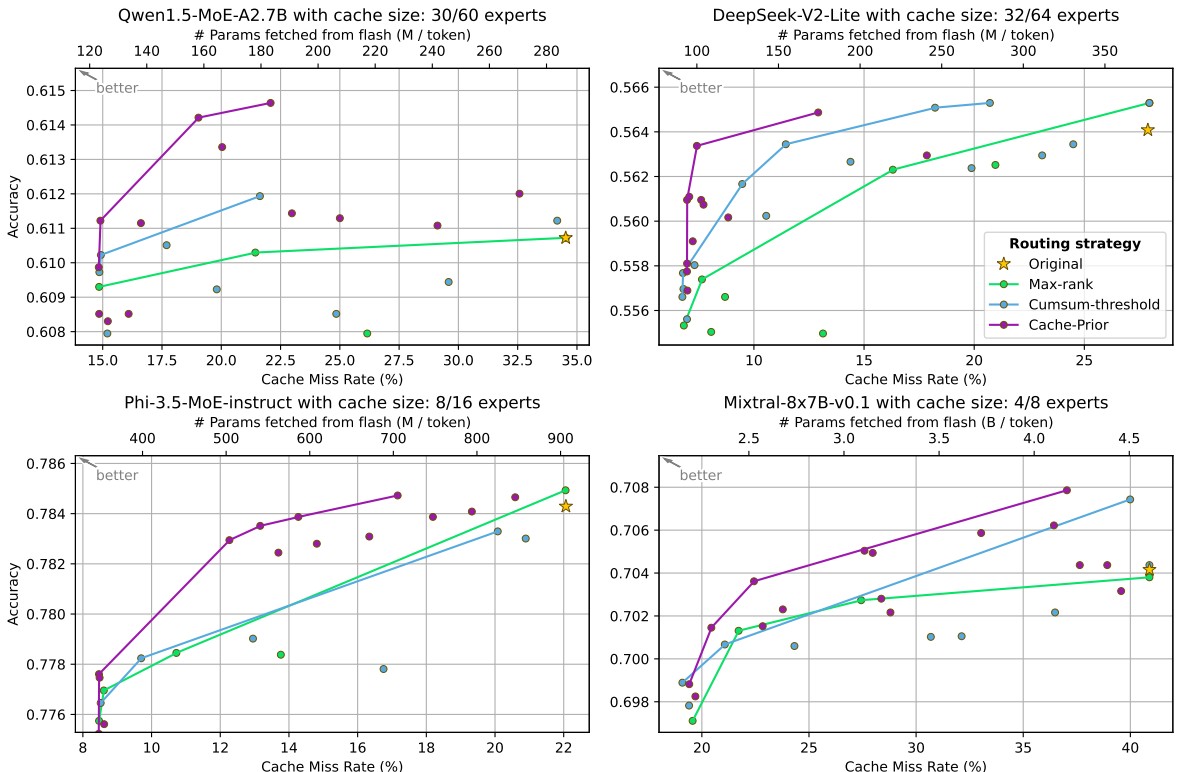

Figure 5: The trade-off between MMLU (5 shots) task accuracy and cache miss rate. For each method, points along the curve form the Pareto front, showcasing the best achievable accuracy for a given cache miss rate.

MoE-A2.7B (Qwen Team, 2024), Phi-3.5-MoE (Abdin et al., 2024), and Mixtral-8x7B (Jiang et al., 2024), with architectural details summarized in Table 1.

## 4.1 Experimental Setup

We conduct experiments on three tasks: language modeling (using the WikiText-2-raw-v1 dataset), MMLU, and GSM8K. For WikiText, we report perplexity and cache miss rate, while for MMLU and GSM8K, we report accuracy and cache miss rate. The MMLU dataset consists of multiple-choice questions across 57 subjects, and GSM8K evaluates multi-step reasoning for math problems. For all experiments, the cache miss rate is computed using the Least Recently Used (LRU) eviction policy, unless stated otherwise in ablations.

To assess the trade-offs between model performance and cache miss rate, we vary DRAM cache size limits and sweep hyperparameters for each method to generate Pareto fronts. All reported results in the trade-off plots are point estimates, obtained from a single pass over the entire dataset. As our algorithm and baselines are deterministic, repeated runs yield identical results. Each plot also includes the performance of the "Original Routing" baseline, where no re-ranking strategy is applied, maintaining accuracy without trade-offs. In contrast, our cache-aware routing methods balance accuracy with cache utilization.

## 4.2 Implementation Details

All routing strategies in this work use the LRU eviction policy for expert removal, where we impose an eviction order by removing experts with higher router weights first, as MoE layers select top-$K$ experts in parallel. For the pruning baseline, experts ranked $\geq h$ (with $1 < h \leq k$) are discarded, and the cache miss rate is normalized using $k$ for a fair comparison with other methods.

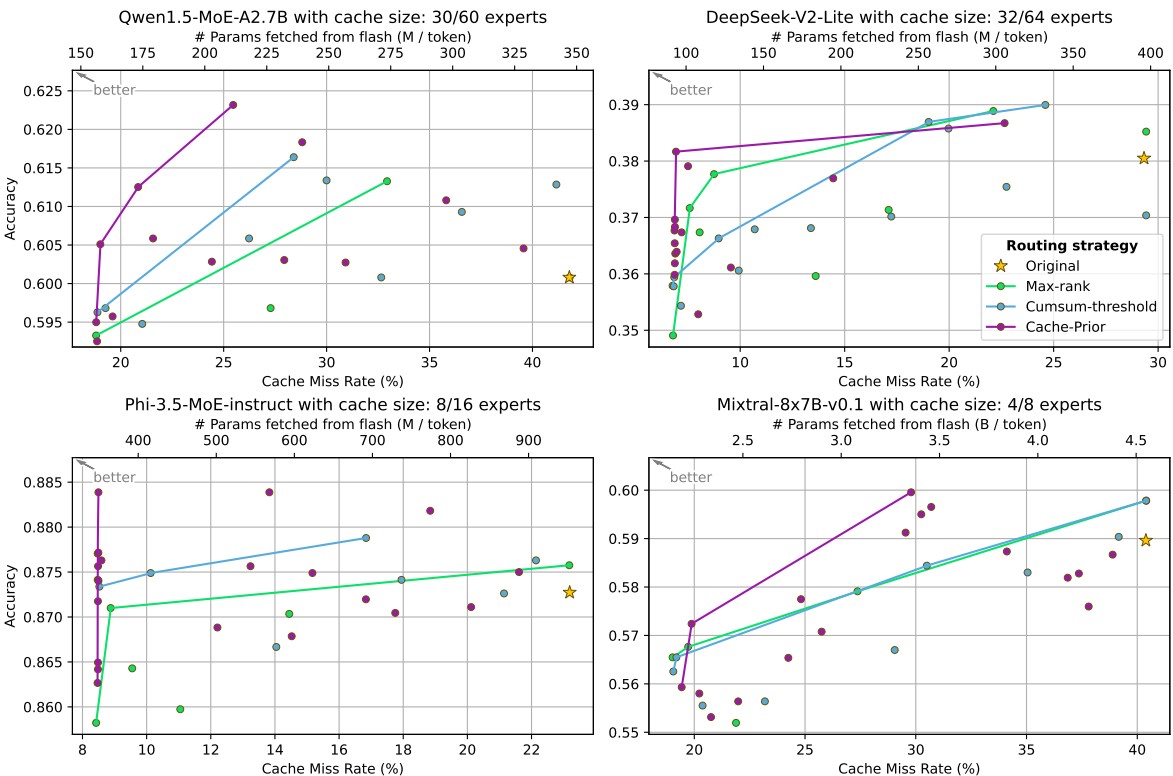

Figure 6: The trade-off between GSM8K (8 shots) task accuracy and cache miss rate. For each method, points along the curve form the Pareto front, showcasing the best achievable accuracy for a given cache miss rate.

Each cache-aware routing strategy has a hyperparameter to balance cache miss rate and task performance. We use the following values to generate Pareto curves: Pruning and Max-Rank use $0, 1, ..., K$, while Cumulative Sum Thresholding and Cache-Prior use 50 equidistant points in $[0, 1]$. For guaranteed top-$J$ loading, we set $J = 1$ for Mixtral and Phi-MoE models and $J = 2$ for the granular Qwen-MoE and DeepSeek-MoE architectures.

For dataset preprocessing, we concatenate WikiText text into a single blob, split by "nn", and chunk it into context lengths of 1024. For MMLU and GSM8K, we apply a few-shot approach (5 shots for MMLU and 8 shots with chain-of-thought for GSM8K). Our method is applied to the entire sequence for WikiText and MMLU, and only during autoregressive generation for GSM8K.

## 4.3 Language Modeling Evaluation

We start with assessing the effect of our proposed cache-aware routing method on the language modeling capabilities of MoE models. Figure 4 presents trade-off curves for perplexity versus cache miss rate of our Cache-Prior approach compared to the Pruning, Max-Rank, and Cumsum-threshold methods with a cache size equal to half the number of experts. The results show that our approach consistently outperforms all baselines across all ranges of perplexity and cache miss rate. Among the baseline methods, Pruning performs the worst, indicating that cache-informed expert replacement is essential for maintaining accuracy. Max-Rank consistently surpasses Pruning, while Cumsum-Threshold consistently outperforms Max-Rank. Ultimately, our Cache-Prior method Pareto-dominates all other approaches.

We also evaluate language modeling performance with only a quarter of experts cached, as shown in Figure 14 in Appendix D. Our method's robustness to different cache sizes makes it appealing for deployment across a range of devices.

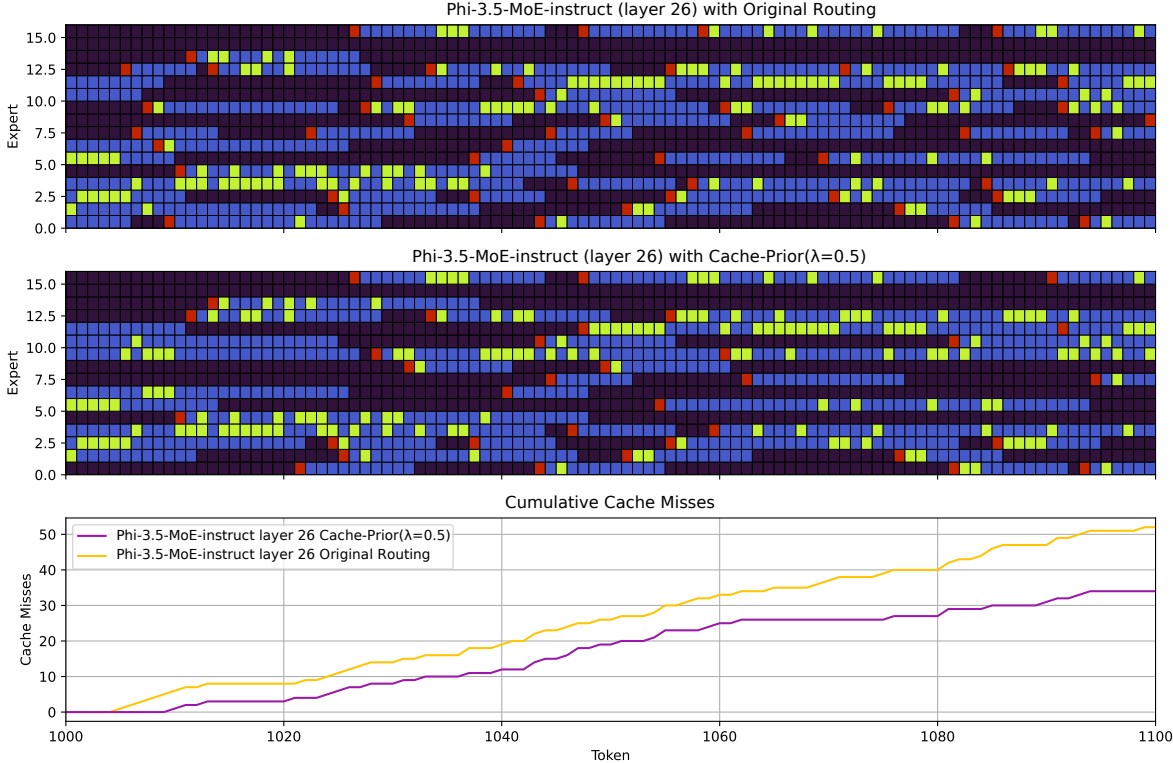

Figure 7: Expert selection during token generation on a GSM8K sample (green: cache hit, red: cache miss, blue: in cache).

## 4.4 Downstream Tasks Evaluation

Figure 5 and Figure 6 present results on the MMLU and GSM8K benchmarks, respectively. Our Cache-Prior method's Pareto curve consistently outperforms other methods, often showing significant cache miss rate reductions with no accuracy loss compared to original routing. Increasing the cache bias $\lambda$ improves cache miss rates until a minimum is reached due to the top-$J$ selection guarantee. We also observe that GSM8K has noisier accuracy results, as can be expected for a generative task. Importantly, while accuracy may fluctuate, adjustments to the hyperparameters for each routing method —namely $M$, $p$, and $\lambda$— consistently leads to a predictable effect on cache miss rates.

Interestingly, we observe that improved cache consistency (reduced miss rate) can lead to increased accuracy on both benchmarks. We hypothesize this may stem from an implicit regularization effect, where greater temporal consistency in expert selection stabilizes predictions, although other factors likely contribute.

## 4.5 On-Device Deployment

To evaluate the effectiveness of our cache-aware routing technique in real-world scenarios, we deployed the Qwen1.5-MoE-A2.7B model with our cache-aware routing on two mobile devices (12GB and 16GB RAM) equipped with Qualcomm Snapdragon® processors running Android 14. Using llama-cpp (Gerganov, 2023) for CPU-based deployment, we modified the implementation to add both LRU caching for experts and our Cache-Prior algorithm. Additionally, we enabled memory locking (mlock) to prevent the Android OS from offloading expert weights from memory.

We tested our approach in two settings: (1) deploying a 4-bit quantized model on a device with 12GB RAM and (2) deploying an 8-bit quantized model on a device with 16GB RAM. In both cases, our method was

---

Snapdragon branded products are products of Qualcomm Technologies, Inc. and/or its subsidiaries.

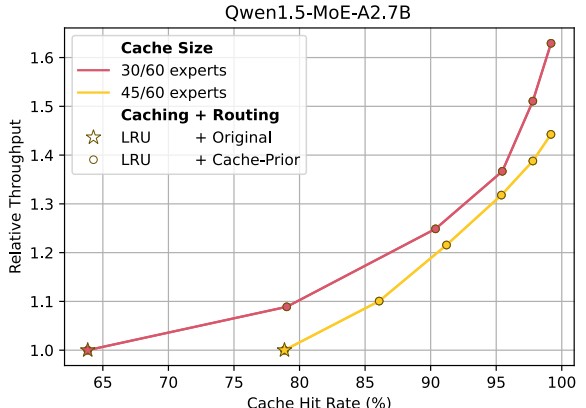 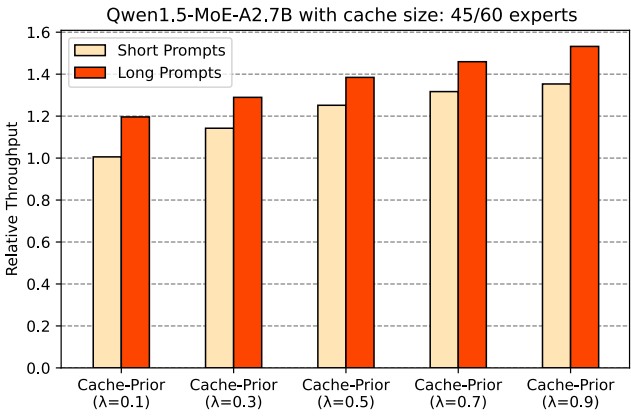

Figure 8: of cache hit rate and prompt length on token generation throughput for the 4-bit quantized Qwen1.5-MoE-A2.7B model on the 16GB device. (Left) Relationship between cache hit rate and relative throughput (normalized to the LRU baseline of 1.0 for each cache size) for cache sizes of 30 and 45 experts (out of 60), across varying $\lambda$ values $0.1 - 0.9$. (Right) Influence of prompt length on relative throughput across varying $\lambda$ values, with a cache size of 45 experts.

applied only during autoregressive generation, allowing the model to benefit from optimizations designed for prompt processing.

In the first setting, we reserved 2GB of memory on the 12GB device, limiting available memory to 10GB (shared with the OS). Here, the cache size was limited to 30 experts per layer (out of 60). In the second setting, the 16GB device hosted the 8-bit quantized Qwen Model with a cache size limit of 45 experts per layer. To determine these optimal cache sizes for each setting, we initially deployed the model on both devices using the LRU caching policy across various cache sizes. Results, shown in Figure 13, indicate that exceeding a certain cache size reduces available memory, causing the OS to offload uncached components (e.g., KV-cache, activations) for each token, requiring reloading them from flash which significantly slows down inference. We used the highest LRU throughput as a reference point (1x) to demonstrate the speedup from our method independently of wall clock variations due to implementation specifics.

The quantitative results for both settings are presented in Figure 1 (Right). Box plots, computed over 10 runs per experiment, show median, minimum, and maximum values. Our approach provides over a 2× speedup on-device. Additionally, qualitative results in Table 3 in the Appendix demonstrate that this speedup has minimal impact on the quality of generated text.

To quantify the relationship between cache hit rate and token generation throughput, we conducted an experiment, deploying the 4-bit quantized Qwen model on the 16GB device and performed a sweep over the Cache-Prior trade-off parameter $\lambda \in \{0.1, 0.3, 0.5, 0.7, 0.9\}$. A random subset of the MMLU dataset, comprising both short (40-60 tokens) and long (300-400 tokens) prompts, was used for these measurements.

Figure 8 (left) illustrates the relation between cache hit rate and relative token generation throughput for two cache configurations: 30 experts and 45 experts cached per layer (out of 60 experts). For each configuration, the throughput achieved with standard LRU caching and original routing (equivalent to $\lambda = 0$ where no expert swapping occurs) serves as the baseline (relative throughput of 1.0). The results demonstrate a near-linear positive relationship: as $\lambda$ increases, the cache hit rate improves, leading to a corresponding increase in throughput. This observed trend is logical as cache-hit rate linearly correlates with the number of flash reads. As loading from flash is the major performance bottleneck, the number of flash reads should correlate roughly linearly with the throughput.

Furthermore, Figure 8 (right) examines the influence of prompt length on throughput for a cache size of 45 experts. Across all values of $\lambda$, longer input prompts yield higher throughput (Similar trend is observed for the cache size of 30 in Appendix F). This is likely attributable to the amortization of initial overheads (such

as model loading and initial cache fills) over a greater number of generated tokens, and potentially more stable expert usage patterns for longer sequences.

## 4.6 Temporal Consistency of Expert Selections

Figure 7 shows a qualitative visualization of cache hits/misses for a GSM8K sample (tokens 1000-1100), comparing original routing and Cache-Prior ($\lambda$=0.5) methods, with cache miss rates of $\sim$23% and $\sim$15%, respectively, for Phi-3.5-MoE-instruct. It is evident that the Cache-Prior method results in fewer cache misses and longer cache lifetimes compared to the original routing baseline.

These observations are further supported by Table 2, which presents average cache miss rates and cache lifetimes (the average number of time steps an expert remains in memory) for the Wikitext dataset. Longer cache lifetimes indicate better memory efficiency.

Using the original routing, each expert in Qwen and DeepSeek models stays in memory for only 22 tokens. With our method, experts remain in memory for $5 - 9\times$ longer, reducing weight loading from Flash and improving throughput. MoE models with fine-grained implementation (Ludziejewski et al., 2024)—where the hidden dimension of each expert is smaller than a standard feed-forward layer (e.g., Qwen-MoE and DeepSeek-MoE)—are more cache-friendly compared to conventional MoE architectures like Phi-MoE and Mixtral.

| Model | Cache Size | Routing | Lifetime | Miss Rate |
|-------|-----------|---------|----------|-----------|
| Qwen1.5-MoE | 30 / 60 | Original | 26 ($\pm$31) | 35% |
| | | Cache-Prior | 58 ($\pm$67) | 16% |
| DeepSeek-V2 | 32 / 64 | Original | 19 ($\pm$17) | 28% |
| | | Cache-Prior | 76 ($\pm$90) | 7% |
| Phi-3.5-MoE | 8 / 16 | Original | 22 ($\pm$31) | 22% |
| | | Cache-Prior | 55 ($\pm$94) | 9% |
| Mixtral-8x7B | 4 / 8 | Original | 5 ($\pm$4) | 40% |
| | | Cache-Prior | 10 ($\pm$10) | 21% |

Table 2: Cache sizes (number of experts that fit in cache / total number of experts in model), average cache lifetimes (in number of tokens), and cache miss rates for LRU and prior caching strategies with $\lambda = 0.5$ on the Wikitext validation set.

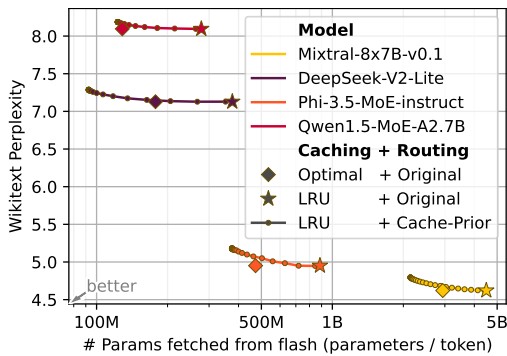

Figure 9: Tradeoff between language modeling performance and the number of parameters loaded from Flash memory . "Optimal" represents the oracle cache eviction bound.

## 4.7 Impact of MoE Design Choices on Caching Performance

While retraining MoE models with varying design choices is beyond this paper's scope, our experiments offer insights into how architectural choices in pre-trained MoEs affect deployment-time caching. The evaluated MoE models (Table 1) represent diverse architectures, differing in granularity, routing mechanisms, expansion rates, and the use of shared experts.

**Granularity** Our results suggest that model granularity (Ludziejewski et al., 2024) influences cache efficiency, particularly when combined with our cache-prior reranking method. We observe that granular MoEs (Qwen-MoE, DeepSeek-MoE) tend to be more resilient to the approximate routing introduced by expert swapping compared to non-granular models like Mixtral. For instance, while Qwen-MoE and Mixtral exhibit comparable initial LRU cache miss rates (35% and 40%, respectively) in Figure 4, our cache-prior method halves Qwen-MoE's miss rate with only a 0.5% perplexity increase. Achieving a similar miss rate reduction for Mixtral incurs a much higher perplexity penalty (2.9%). DeepSeek-MoE demonstrates even greater resilience, halving its baseline miss rate with a negligible perplexity increase ( 0.1%).

**Expansion Rate**   The expansion rate, defined as the ratio of activated expert parameters to the total number of expert parameters (Ludziejewski et al., 2024), also appears correlated with cache performance. We observe that models with lower expansion rates generally exhibit lower baseline cache miss rates. Specifically, Phi-MoE, Qwen-MoE, and DeepSeek-MoE (all with an expansion rate of 0.125) tend to show better cache characteristics under LRU compared to Mixtral (expansion rate of 0.25).

**Top-K**   We observed no clear link between top-$K$ and overall cache-friendliness (e.g., Mixtral vs. Phi, both $k = 2$). However, models with smaller $k$ (Mixtral, Phi) are more sensitive to top-1 expert swaps and benefit more from guaranteeing top-1 expert loading (Figure 2 and Figure 4). In contrast, Qwen ($k = 4$) and DeepSeek ($k = 6$) show greater resilience to expert swapping, although their use of shared experts might also contribute to this behavior.

### 4.8   Ablation Experiments

In this section, we perform ablation experiments on the WikiText dataset to demonstrate the general applicability of our method across a range of cache sizes, compare its performance to the theoretical bounds of the optimal cache eviction policy, and justify our hyperparameter choices.

**Optimal Cache Eviction**   As a theoretical upper bound for cache policy, we consider Belady's algorithm (Belady, 1966), which evicts the expert that will be needed farthest in the future. This lossless policy requires perfect foresight of future router decisions, making it unattainable in practice but useful as an upper bound on cache hit rate. Figure 9 shows that while Belady's algorithm roughly halves cache miss rates compared to LRU.

Conventional caching methods (e.g., LRU, Belady's optimal) are lossless and thus cannot surpass the theoretical cache hit rate bound set by the model's own routing. In contrast, our approach introduces controlled, lossy, reranking of experts, allowing the model to swap lower-ranked experts for in-cache ones, at the cost of a small, tunable increase in perplexity. This design enables us to trade a negligible amount of accuracy for a substantial gain in cache efficiency, even surpassing the oracle bound. As can be seen in Figure 9, our Cache-Prior method achieves similar or even lower miss rates with only minimal perplexity increases.

**Cache Size**   We investigate whether our findings hold across different cache sizes, beyond our default of half the total expert count. Figure 10 shows results for cache sizes ranging from one expert to all experts, comparing LRU, optimal caching, and our Cache-Prior method (with an LRU cache). Unlike the first two approaches, Cache-Prior introduces a perplexity trade-off controlled by $\lambda$. To standardize this comparison, we select the $\lambda$ value for each model and cache size that minimizes cache miss rate while keeping the perplexity increase within specific thresholds (1%, 5%, and 10%, shown in different shades of purple).

As expected, when the cache size equals $N$, all experts fit in memory, and the miss rate drops to zero for all methods. For the optimal cache baseline, reducing cache size to half the number of experts improves performance over LRU, but the benefit diminishes as cache size shrinks further, becoming negligible for cache sizes $\leq k$.

In contrast, our Cache-Prior method consistently enhances performance, particularly at smaller cache capacities. At the extreme of a cache size of one, even the optimal caching strategy yields only marginal improvements. For models with high values of $k$, our method behaves similarly, as cache reuse still results in $k - 1$ cache misses. However, for models with smaller values of $k = 2$, such as Mixtral and Phi, Cache-Prior significantly improves cache hit rates, achieving up to a 20% reduction in cache misses even at the limit of cache size $= 1$.

Remarkably, with just a 1% increase in perplexity, our Cache-Prior method outperforms the theoretical optimal caching bound for all cache sizes and models, except Phi-3.5-MoE-Instruct. When allowing a 5% increase in perplexity, our method surpasses the optimal bound across all models and cache sizes. This demonstrates the effectiveness of Cache-Prior, as it can exceed the theoretical optimal caching performance with only a minimal trade-off in downstream task performance.

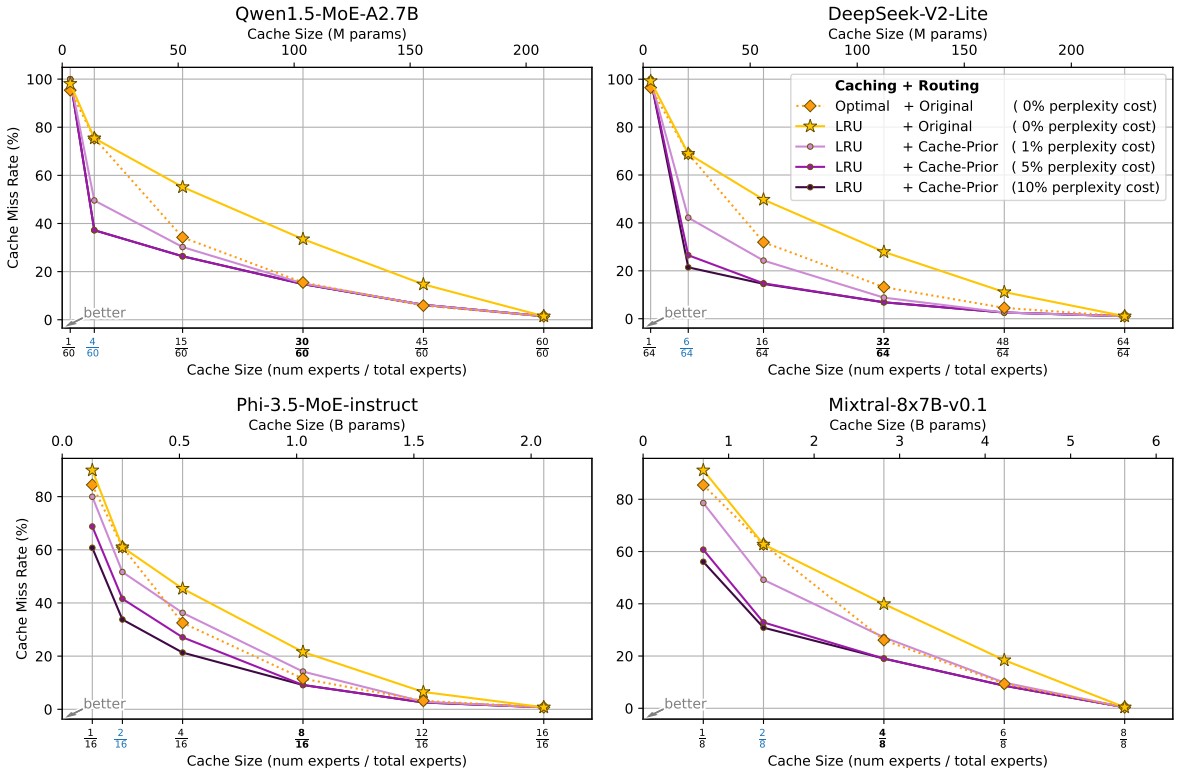

Figure 10: Cache size ablation on Wikitext. Cache size of half of the total number of experts $N$ (highlighted in bold) is used throughout the rest of this paper. Cache size of $k$ is highlighted in blue.

**Impact of Initial Cache State**   A potential concern with cache-aware expert routing is that the initial cache content might bias the model to repeatedly select the same experts, thereby limiting diversity and adaptability in expert selection. To investigate this, Figure 18 in Appendix G visualizes cache hits/misses for a GSM8K sample at the start of prompt encoding, for the original routing and our Cache-Prior ($\lambda = 0.5$ and $\lambda = 0.8$) method. We consider two scenarios: (1) the initial cache is empty and fills as experts are selected during processing of the first few tokens, and (2) the cache is initialized with a random set of experts. We provide more details on this ablation in Appendix G.

Our results show that, for moderate values of $\lambda$ (e.g., 0.5), the initial cache state has minimal long-term impact: after processing a few tokens, the distribution of expert activations and the cache state converge, regardless of initialization. This indicates that the routing mechanism does not remain biased toward the initial cache content, and the method remains robust to different initial cache states without degrading model diversity or generation performance. In practice, the strength of the bias towards cached experts can be controlled by the parameter $\lambda$. When using excessively larger $\lambda$ values (e.g., 0.8), the model may become overly reliant on the cached experts, potentially reducing predictive performance.

**Varying the Number of top-$J$ Experts**   As discussed in Section 4.2, we first select the top-$J$ experts ($J < K$) from the router's normal top-$K$ selection, then complement the remaining experts by favoring those already in the cache, ensuring both critical experts and cache-efficient choices are included. Figure 4 shows the effect of varying $J$ on language modeling performance. While baseline methods like Max-rank and Cumsum-threshold experience a significant performance drop without loading the top-$J$ experts, Cache-Prior is more robust to changes in $J$. The impact on Cache-Prior is model-dependent, with minor improvements in perplexity for models like DeepSeek, Mixtral, and Phi, and a slight decrease for models like QWEN. Given that $J > 0$ benefits baseline methods and produces adequate results for Cache-Prior, we keep it fixed across all methods.

## 5 Related Work

**Expert Caching**   Caching is a common optimization in MoE models, given their high memory requirements. Most prior works rely on a cache with an LRU eviction policy (Eliseev & Mazur, 2023; Yuan et al., 2024b; Yi et al., 2023; Hwang et al., 2024; Huang et al., 2023), while others adopt a least-frequently-used (LFU) eviction policy (Xue et al., 2024a) or evict experts based on more dynamic and custom measures of expert importance (Kamahori et al., 2024; Kong et al., 2023).

These methods manage expert storage and eviction without changing the model's routing decisions or outputs and thus also maintain identical performance. However, this constraint limits their effectiveness, as they cannot change which experts are selected. To establish a performance upper bound, we include an optimal cache baseline that assumes perfect foresight of future router decisions. While this is unattainable in practice, we show that our method can surpass this theoretical bound with only a minimal increase in perplexity. Unlike conventional caching strategies, our approach introduces a controlled trade-off between accuracy and latency, enabling more flexible resource management.

Other lossy caching strategies include Adapt-MoE (Zhong et al., 2024), which dynamically adjusts cache sizes per layer, and HOBBIT (Tang et al., 2024), which decides whether to fully load, skip, or use lower-bitwidth versions of experts based on router confidence. However, both rely on complex prefetching mechanisms. In contrast, our method is much simpler, requiring only a single hyperparameter to balance accuracy and latency effectively.

**Speculative Routing and Weight Prefetching**   Several approaches optimize MoE inference by prefetching expert weights and using speculative routing (Cui et al., 2023; Xue et al., 2024b; Eliseev & Mazur, 2023; Du et al., 2024). These methods improve cache hit rates by predicting which experts will be needed in advance. For example, MoE-Infinity (Xue et al., 2024b) uses look-ahead routing and caches the most frequently used experts. SiDA-MoE employs an offline-trained hash function to reduce expert selection overhead, though at a significant cost to perplexity. Similarly, Eliseev & Mazur (2023) combines look-ahead routing with LRU caching, requiring the pre-loading of multiple experts well in advance to optimize the overlap between data movement and inference calculations. While this approach helps conceal memory transfer costs, it can adversely affect model accuracy. Speculative routing to improve cache hit-rate without loss of accuracy is particularly challenging on memory-constrained devices where token generation is more memory-bound than compute-bound. In contrast, our method enhances routing consistency by exploiting locality priors, achieving minimal increase in perplexity and significant latency improvements. Our approach is orthogonal to speculative routing methods.

**Efficient LLM Inference**   Techniques such as LLM quantization (Dettmers et al., 2022; Frantar et al., 2023; Shao et al., 2024) and compression (Frantar & Alistarh, 2023) can significantly reduce the memory footprint of large language models (LLMs). Recent studies have explored pruning or merging MoE experts based on their utilization to conserve memory (Chen et al., 2022; Li et al., 2024; Lu et al., 2024), though this can result in decreased performance. To facilitate the deployment of off-the-shelf MoEs on devices with limited memory without a model surgery that would change the architecture, previous works have focused on optimizations like offloading parameters to host memory (Alizadeh et al., 2024; Yu et al., 2024; Jung et al., 2023; Rasley et al., 2020). These approaches employ dynamic scheduling strategies to offload sparse parameters while maximizing the overlap between expert loading and computation. For instance, DeepUM (Jung et al., 2023) records CUDA kernel execution patterns to predict which kernel is likely to be executed next. However, MoE model execution patterns can vary significantly depending on the task and context. Our proposed method enhances expert cache hit rates without requiring complex hardware implementations and remains agnostic to the specific task being addressed.

## 6 Conclusion

In this work, we introduced a novel training-free, cache-aware expert routing approach for Mixture of Experts models, designed to improve inference efficiency on memory-constrained hardware. By increasing cache hit

rates through consistent routing, our method significantly reduces latency while preserving model accuracy, achieving up to a 2× speedup over standard LRU caching on mobile devices.

Our experiments demonstrate that models remain robust to expert swapping, particularly when the top-1 expert is preserved. This suggests new possibilities for balancing performance with cache efficiency. To explore this, we introduced three routing strategies, with our Cache-Prior method proving particularly effective. Unlike prior work, which is fundamentally constrained by the optimal oracle cache bound, we show that allowing minimal deviations in expert selection enables us to surpass this theoretical limit while maintaining strong downstream task performance.

We conducted a comprehensive analysis across four state-of-the-art models—DeepSeek-V2-Lite, Qwen1.5-MoE-A2.7B, Phi-3.5-MoE, and Mixtral-8×7B—on three tasks: WikiText, MMLU, and GSM8K. Our Cache-Prior method achieves a substantial reduction in cache miss rates across all models, decreasing them by more than 50%, while incurring only a minimal increase in perplexity of 0.1%–3% on the language modeling task and a negligible accuracy drop of less than 0.1% on downstream tasks. Additionally, our findings indicate that granular MoE models, such as DeepSeek-V2-Lite and Qwen1.5-MoE-A2.7B, are inherently more cache-efficient. This trend suggests a promising future for newer MoE architectures like DeepSeek v3 and Qwen2.5max, which, despite being too large for mobile deployment today, could benefit from their cache-friendly structure.

Our method is highly adaptable, requiring no model retraining, making it suitable for a wide range of hardware configurations. This adaptability reduces deployment costs and facilitates the practical application of MoE models in real-world, resource-limited environments. A key feature of our approach is that it enables explicit control over the trade-off between cache efficiency and predictive performance via the $\lambda$ parameter. While excessively large values of $\lambda$ can introduce a brief, transient bias toward the cache state and may encourage overuse of cached experts, this trade-off is tunable and can be adjusted to suit specific deployment needs. In practice, moderate values of $\lambda$ provide substantial cache efficiency gains with minimal impact on accuracy.

In summary, our cache-aware expert routing approach not only enhances inference efficiency but also opens new avenues for deploying large MoE models in resource-limited environments. This work paves the way for more practical and cost-effective applications of MoE models, ensuring their scalability and adaptability in diverse hardware configurations.

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

# Appendix

## A    Optimal Expert Selection

To investigate the accuracy of the router's expert predictions, we conduct an experiment on Wikitext using Mixtral-8x7B-v0.1. Iterating over layers, we fix the top-1 expert and systematically test all possible choices for the second expert, finding the one that minimizes perplexity. The results, shown in Figure 11, reveal that the router selects the optimal top-2 expert only 28% of the time on average. Although deeper layers improve this accuracy, the best-performing router only correctly predicts the top-2 expert in just 38% of cases. This suggests that while selecting the highest-ranked experts is important, router predictions are often suboptimal, leaving room for flexibility in expert selection without significantly impacting performance.

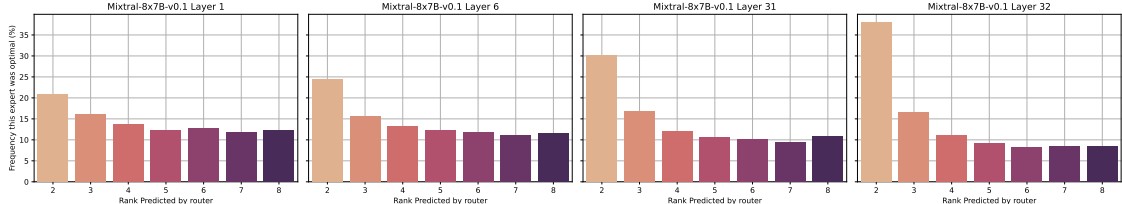

Figure 11: Agreement between router predictions and optimal expert selection across different layers of Mixtral-8x7B-v0.1. The y-axis represents the frequency with which a predicted expert was actually the optimal choice based on Wikitext perplexity.

## B    Intuitive Explanation of the Max Rank and Cumulative Sum Threshold Baselines

To provide further intuition, the Max Rank approach can be understood as follows. Rather than always selecting the top-$K$ experts with the highest router probabilities, we consider a slightly larger pool of candidate experts, specifically, those ranked in the top $M$ by the router. Among these, we give priority to experts that are already present in the cache, promoting them to the front of the ranking. This increases the likelihood that the selected experts are already loaded in fast memory, reducing cache misses and improving inference efficiency. However, to avoid degrading model performance, we do not promote cached experts that fall outside the top $M$ router ranks, ensuring that only reasonably probable experts are considered. In effect, this method balances the trade-off between cache efficiency and predictive accuracy by flexibly reordering the selection within a controlled rank window.

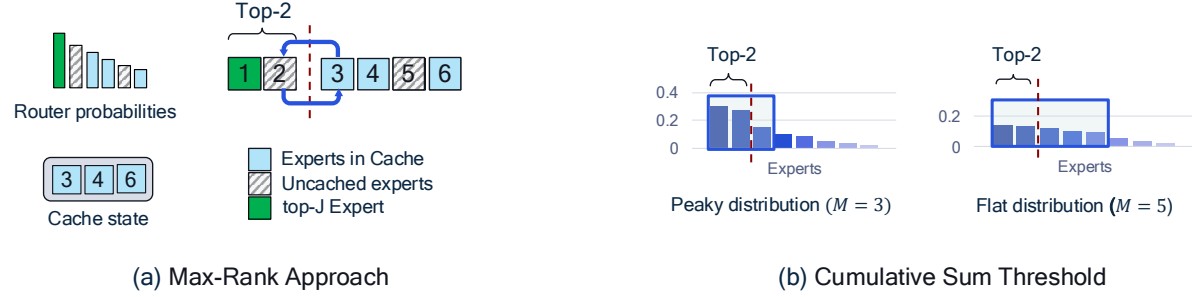

(a) Max-Rank Approach                        (b) Cumulative Sum Threshold

Figure 12: Illustration of the Max Rank and Cumulative Sum Threshold Baselines.

To provide an example, suppose the router produces a ranked list of experts $\mathbf{r} = [E_1, E_2, E_3, E_4, E_5, E_6]$, and the cache currently contains $C = \{E_3, E_4, E_6\}$ (See Figure 12a for an illustration). Let $M = 4$, $K = 2$, and $J = 1$. The top $M$ experts are $[E_1, E_2, E_3, E_4]$. The intersection with the cache yields $[E_3, E_4]$. Applying the promotion operation, we reorder the list to obtain $[E_3, E_4, E_1, E_2, E_5, E_6]$. To ensure the top-J is always included, we move them to the front in order while preserving the rest $[E_1, E_3, E_4, E_2, E_5, E_6]$. The top-$K$ experts selected for this token would then be $E_1$ and $E_3$.

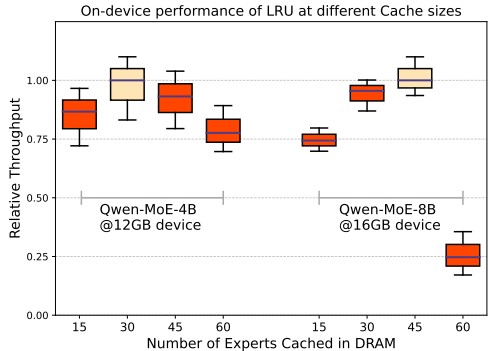

Figure 13: Throughput of the 4-bit and 8-bit quantized Qwen1.5-MoE models with LRU caching deployed on two mobile devices with 12GB and 16GB available memory, respectively.

The procedure for expert swapping in the Cumulative Sum Threshold approach is based on the same principle, however, the maximum rank $M$ is dynamically determined by thresholding the cumulative sum of sorted router probabilities. As can be seen in Figure 12b, for flat distributions, $M$ is larger and for peaky distributions, $M$ takes smaller values.

## C  On-Device LRU Throughput

In Figure 13 we show on-device performance for various cache sizes. Similar to Section 4.5, we use best LRU performance as a reference point of $1\times$. As can be seen, increasing cache size beyond 30 in case of 4-bit quantized model on 12GB device (left) and beyond 45 in case of 8-bit quantized model on 16GB device (right) leads to decrease in performance. This is due to the fact that increasing cache size reduces available memory, causing the OS to offload uncached components (e.g., KV-cache, activations) for each token, requiring reloading them from flash which significantly slows down inference.

Given these results, we use same cache size of 30 for 4-bit and 45 for 8-bit quantized models in experiments in Section 4.5.

## D  Smaller Cache Sizes

Throughout all the experiments in the paper, we use a cache size that is $\frac{1}{2}$ times the total number of experts. In Figure 14 we demonstrate performance of our approach for models with cache size equal to $\frac{1}{4}$ of the total amount. Our Cache-Prior method is outperforming all other methods across the board. Note that it is doing so without any changes to method itself making it appealing for final user who might want to deploy it across a range of devices. An aggregated overview of additional cache sizes can be found in Figure 10 in the main text.

**Logit Range Estimation**  In our method, the scaling factor $\lambda$ is multiplied by a running average of the logits range $\Delta_{\text{avg}}$, as shown in Equation 9. This estimate of the logit range is dynamic, depending on model and input. To improve the estimate, we could alternatively calculate $\Delta_{\text{avg}}$ over a calibration dataset. We tested this by estimating the range over the entire Wikitext training subset. Results in Figure 15 show that our running average performs comparably to estimates based on the full dataset, while also offering robustness to out-of-domain data where logit ranges may differ from general text data (e.g., for a coding task).

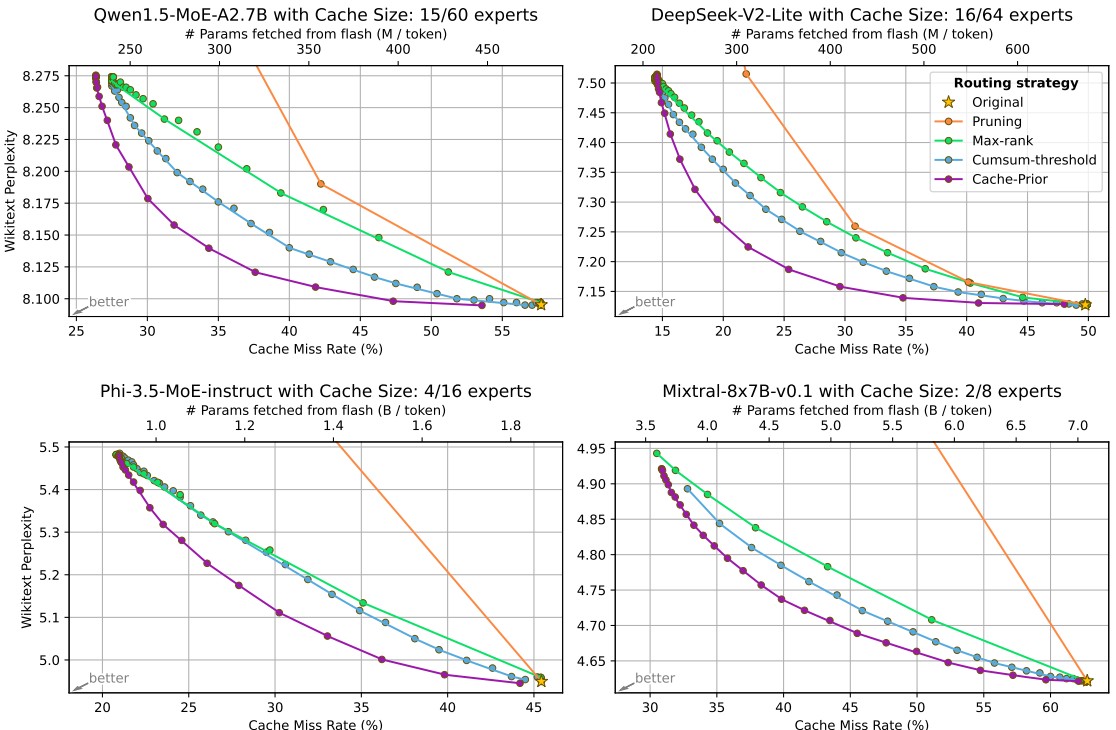

Figure 14: Wikitext Perplexity results for cache size that is one quarter the amount of experts.

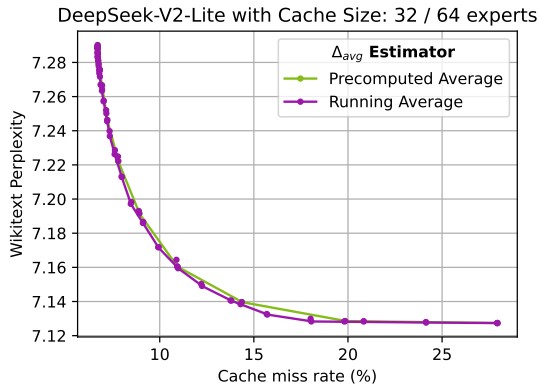

Figure 15: Effect of different strategies for estimating the logit range.

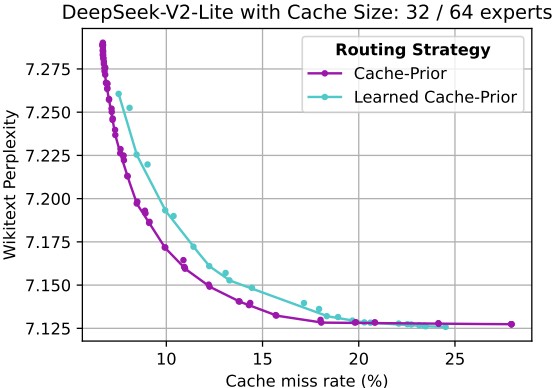

Figure 16: Performance of our learned Cache-Prior strategy on Wikitext.

# E   Learned Cache-Prior

In Section 3.3 we present our training-free Cache-Aware routing method. As shown throughout the paper, the presented method is very strong and competitive across a range of scenarios.

However, it is also possible to make the proposed method learnable. To do so, we propose learning the additive cache-prior term using a small cache MLP layer conditioned on the cache state. Specifically, we implement this by employing a two-layer MLP that takes both the cache state and router logits as input, outputting a single bias vector. This bias is then added to the router logits and used for the next step. The loss function is optimized on the softmax outputs. We force weights assigned to experts that are in the cache

but not in top-$K$ to move closer to one while penalizing weights for those that are in top-$K$ but not in the cache to move closer to zero.

A downside of learnable cache-priors is that we are required to train many independent cache MLPs to obtain trade-off points between cache hit rate and model accuracy. We conducted an experiment to verify the effectiveness of learning this bias but did not observe notable improvements. This optimization is challenging and the results were close to the cache-prior performance without outperforming it. Moreover, this approach would be less beneficial in settings with multiple deployment targets. Therefore, the training-free nature of our approach proves very advantageous for real-world applications.

Given the increased training expense, comparable performance and limited applicability, we skip this approach for now, leaving it for future work.

## F Impact of Prompt Length on Throughput

Figure 17 shows the influence of prompt length (short: 40-60 tokens; long: 300-400 tokens) on throughput for the Qwen1.5-MoE-A2.7 model deployed with a cache size of 30 experts. Across nearly all values of $\lambda$, longer input prompts yield higher throughput.

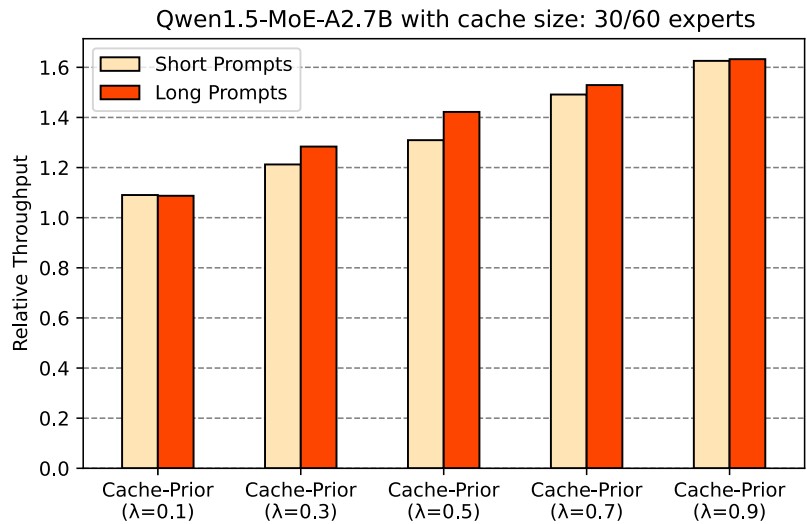

Figure 17: Influence of prompt length on relative throughput across varying $\lambda$ values, for the Qwen1.5-MoE-A2.7 running on the 16GB device with a cache size of 30 experts.

## G Impact of Initial Cache State

In this section, we qualitatively assess how the initial cache state affects expert selection. Figure 18 visualizes cache hits and misses for a GSM8K sample during the prompt encoding phase for the Phi-3.5-MoE-instruct model. We compare both the original routing and Cache-Prior methods for a cache size of 8 experts across four scenarios: (1) original routing with an empty cache, (2) Cache-Prior ($\lambda = 0.5$) with an empty cache, (3) Cache-Prior ($\lambda = 0.5$) with a randomly initialized cache, and (4) Cache-Prior ($\lambda = 0.8$) with the same random cache state as in *(3)*. This setup allows us to examine the influence of both the initial cache content and the $\lambda$ parameter on expert selection. In all these setting, we set $J = 1$, for guaranteed top-1 expert loading.

Our results show that, for moderately large values of $\lambda$ (e.g., 0.5), the initial cache state has minimal long-term effect: after processing a few tokens, the distribution of expert activations and the cache state converge, regardless of how the cache was initialized. This suggests that the routing mechanism is not persistently biased toward the initial cache content. However, with excessively larger $\lambda$ (e.g., 0.8), we observe that the model tends to overly reuse experts in the cache, which may negatively impact predictive performance.

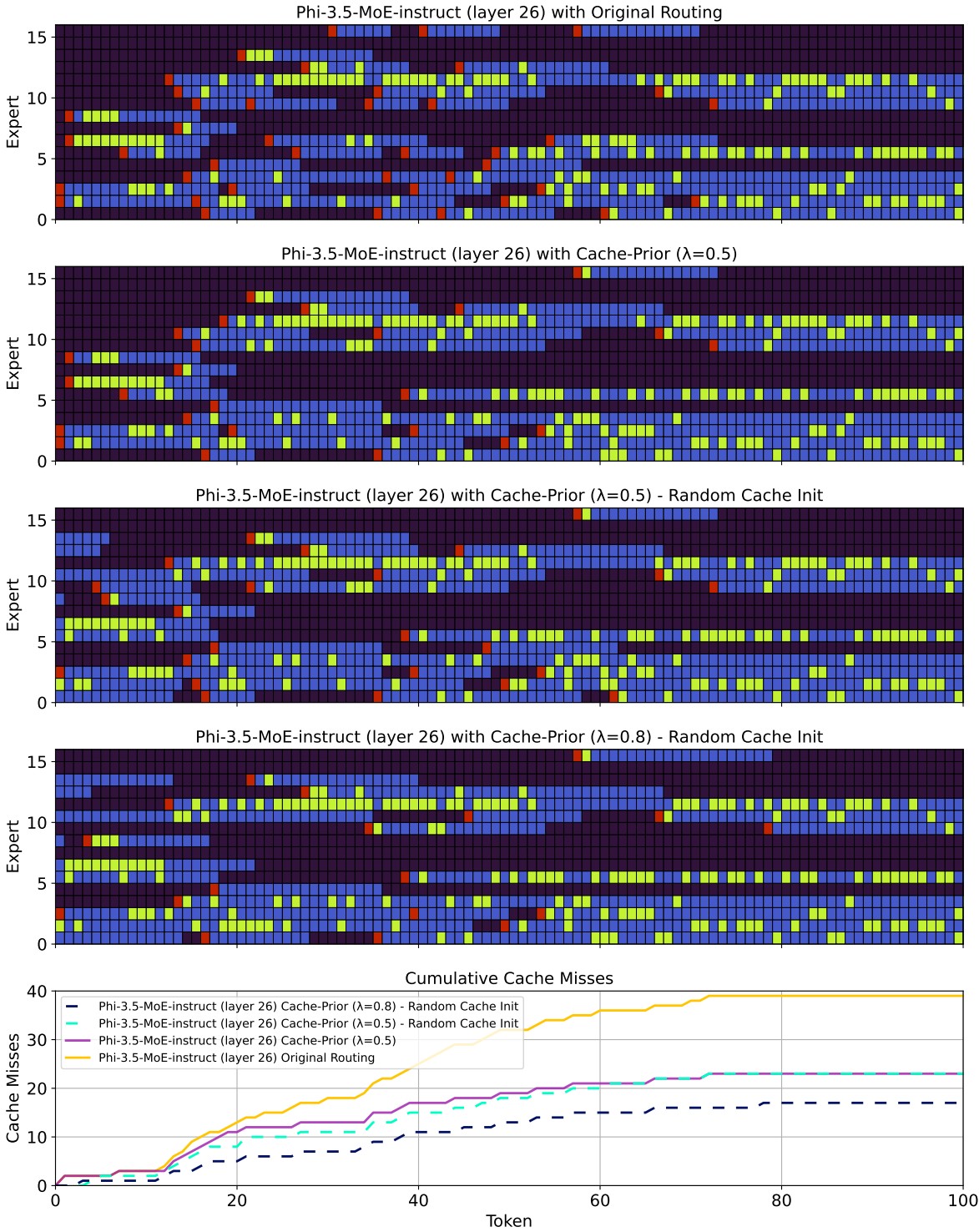

Figure 18: Expert selection during token generation on a GSM8K sample (green: cache hit, red: cache miss, blue: in cache).

## H  Qualitative On-Device Results

Table 3 presents sample text from our on-device autoregressive text generation experiment. It is evident that incorporating our Cache-Prior method has minimal impact on the quality of the generated results. For more details, refer to Section 4.5 of the main text.

| Model | Q4, Cache Size = 45 | Q8, Cache Size = 30 |
|---|---|---|
| Prompt | Write a short poem inspired by "Numb" by Linkin Park but about Mixture-of-Expert models. | Building a website can be done in 10 simple steps: Step 1: |
| LRU | The model is a type of artificial intelligence that uses multiple machine learning algorithms to make predictions or decisions. It's designed to make better decisions faster than an expert, but it can also make mistakes due to the complexity of combining the decisions of different algo-

In the realm of data and code,
Lies a model so bold and brave,
A mixture of experts, it's its name,
rithms. Working together, making decisions, it thrives.

With algorithms galore, it's a sight to behold,
Predicting and deciding, with speed and force,
Majestic in its complexity,
A marvel of modern science. | Get a website domain name. This is the address where your website will live. You can choose a name that describes your business or brand. It's best to choose a name that's easy for people to remember and spell.
Step 2: Find a web hosting service. A web hosting service is where your website files will be stored. There are many different types of web hosting services to choose from, including shared hosting, VPS hosting, and dedicated hosting. You'll want to choose a hosting |
| Prior, $\lambda = 0.2$ | The model predicts the probability distribution of the next word (y_n) as a function of the previous word (y_{n-1}) and the hidden state (h_{n-1})

(Verse 1)
In the depths of data streams, a secret blend of math, Where words and words and words are churned into a model, The Mixture-of-Experts reigns, a wiser way to guess, The next word to fill the verse, the rhythm of the next verse.

(Chorus)
Numb to the noise, numb to the noise,
Guided by an algorithm, the words flow like a guide,
Numb to the noise, numb to the noise,
Predicting the future with a pattern we can't annoy. | Get a website name (the web address). You need to choose a domain name (or URL) for your website, and register it with a domain registrar. This is the address that people will use to find your website on the internet, such aswww.yourdomain.com.

Step 2: Get web hosting for your website. This is the place where your website will live on the internet, and where yourwebsite files will be stored. The most popular option is to use a hosting company, |
| Prior, $\lambda = 0.8$ | The first part of the English language Wikipedia choose Mixture-of-Expert models as the 2018 word. Mixture-of-Expert models use multiple neural networks as expert solutions. I'm not a poet, but here's my attempt at a very loose and free poetry style translation of the given meaning.
A model so complex, Experts gather from near and far The experts of yore, to bring their expertise, their might A model to optimize, to help us all survive From AI to the rescue, it's transformation we'll rejoice In the mind of Chester and Patel, Expert insights to bestow, Insights to empower, | Find a Domain Name Before you start building your website, you will need to choose a domain name (the name of your website). Your domain name is the address that people will use to reach your website, so you want to make sure it's easy to remember and spell and related to what your website is all about. For example, if you're building a website for your dog grooming business, your domain name could be.

Step 2: Choose a Hosting Provider
Now that you have your |

Table 3: Qualitative results of caching performance. We generate 100 tokens for each prompt.

