# OpenReview forum: "Mixture of Cache-Conditional Experts for Efficient Mobile Device Inference"
_TMLR — Accepted by TMLR_

### Review · Reviewer_XijN · 2025-03-26

**Summary Of Contributions:**

This paper presents a cache-aware efficient MoE algorithm that reduces cache miss rate. The authors demonstrate that the models' performance exhibits low sensitivity to the selection of low-weighted experts, and build a hardware-aware algorithm on top of that. They evaluate the method and prove its efficiency on various benchmarks. They also test it on mobile devices, showing the flexibility and cost-effectiveness of the method.

**Audience:**

Yes

**Claims And Evidence:**

Yes

**Requested Changes:**

1. Profiling results on how much the cache hit rate affects the latency.
2. Discuss the possible negative effects of performing cache-aware expert selection.
3. More results on latency or throughputs improvement.

**Strengths And Weaknesses:**

Strengths:

1. The paper identifies an interesting trend where less important experts do not affect the performance very much, giving opportunity for expert pruning.
2. The method is cost effecitive and does not require significant architecture change or training.
3. The performance results look promising for both cache hit rate and algorithm accuracy.
4. Good ablations on different settings and optimal solution.

Weaknesses:

1. How significant is cache hit rate to the latency of the model serving? Is it the bottleneck?
2. Will the selection of experts in cache cause bias in the output? Will the same set of cache being always selected? How is the performance affected by the initial set of experts in the cache? In short, what are the limitations of the current method?
3. Lack of experiment results on the materialized benefits of increased cache hit rate, like latency and throughputs.

---

> ### Author Response · Authors · 2025-05-10
>
> We thank the reviewer for their thorough evaluation and constructive feedback, as well as recognizing the key strengths of our work, including the novel observation on expert sensitivity, the method’s cost-effectiveness, the promising performance results, and our thorough ablations. We also appreciate the insightful comments and suggestions for further experiments, which have helped us improve the paper.
>
> We have carefully considered all your comments and suggestions and made corresponding improvements in the revised manuscript, which we have highlighted in blue for your convenience.
>
> **Requested Changes 1 & 3: Impact of cache hit rate on latency & throughput, and more results on latency or throughput improvement**
>
> We agree with the reviewer that demonstrating the direct impact of cache hit rate on latency and throughput is crucial. To address this, we have conducted new on-device experiments and added new results to the revised manuscript.
>
> #### **Relation between cache hit rate and throughput**
> To quantify the relationship between cache hit rate and token generation throughput, we conducted an experiment, deploying the 4-bit quantized Qwen model on the 16GB device and performed a sweep over the Cache-Prior parameter $\lambda \in {0.1, 0.3, 0.5, 0.7, 0.9}$. A random subset of MMLU, comprising both short (40-60 tokens) and long (300-400 tokens) prompts, was used for these measurements. For each experiment, we report the average performance over 10 runs.
>
> The newly added *Fig 8* (left) illustrates the relation between cache hit rate and relative token generation throughput for two cache settings: 30 and 45 experts cached per layer (out of 60 experts). For each setting, the throughput achieved with standard LRU caching and original routing (equivalent to $\lambda=0$ where no expert swapping occurs) serves as the baseline (relative throughput of 1.0). The results demonstrate a near-linear positive relationship: as $\lambda$ increases, the cache hit rate improves, leading to a corresponding increase in throughput. This observed trend is logical as cache-hit rate linearly correlates with the number of flash reads. As loading from flash is the major performance bottleneck, the number of flash reads should correlate roughly linearly with the throughput.
>
> #### **Influence of prompt length on throughput**
> *Fig 8* (right) examines the influence of prompt length on throughput for a cache size of 45 experts. Across all values of $\lambda$, longer input prompts yield higher throughput (Similar trend observed for the cache size 30 in Appendix F). This can potentially be attributed to more stable expert usage patterns for longer sequences.
>
> **Bias caused by cache state & discussion on potential Limitation**
> We have performed a new ablation study to investigate potential biases and the impact of the initial cache state, and expanded our discussion on the limitations as follows:
>
> > A potential concern with cache-aware expert routing is that the initial cache content might bias the model to repeatedly select the same experts, thereby limiting diversity and adaptability in expert selection. To investigate this, *Fig 18* in Appendix G, now visualizes cache hits/misses for a GSM8K sample at the start of prompt encoding, comparing the original routing with our Cache-Prior method (using $\lambda=0.5$ and $\lambda=0.8$). We considered two scenarios: (1) an initially empty cache that fills as experts are selected, and (2) a cache initialized with a random set of experts. More details on this ablation are provided in Appendix G.
>
> > Our findings show that for moderate $\lambda$ values (e.g., 0.5), the initial cache state has minimal long-term impact. After processing a few tokens, the distribution of expert activations and the cache state converge, regardless of initialization. This indicates that the routing mechanism does not remain persistently biased by the initial cache content, and the method remains robust to different initial cache states without degrading model diversity or generation performance.
>
> However, the strength of the bias towards cached experts is controllable via the $\lambda$ parameter. As discussed in Appendix G and now highlighted as a controllable trade-off in our revised Conclusion (Sec 6), excessively large $\lambda$ values (e.g., 0.8) can indeed lead to an over-reliance on cached experts, potentially impacting predictive performance. We have added the following statement in our conclusion:
>
> > A key feature of our approach is that it enables explicit control over the trade-off between cache efficiency and predictive performance via the $\lambda$ parameter. While excessively large values of $\lambda$ can introduce a brief, transient bias toward the cache state and may encourage overuse of cached experts, this trade-off is tunable and can be adjusted to suit specific deployment needs. In practice, moderate values of $\lambda$ provide substantial cache efficiency gains with minimal impact on accuracy.

---

> > ### Comment · Reviewer_XijN · 2025-05-16
> > **Acknowledgement**
> >
> > Thank you for addressing my concerns. I am happy with the thoughput results and ablation on initial expert.

---

### Review · Reviewer_8bhs · 2025-04-19

**Summary Of Contributions:**

The paper studies approaches to improve the efficiency of Mixture-of-Experts (MoE) deployment on memory-constrained devices, specifically for token generation with batch size equal to 1. The paper observes that the experts beyond top-1 or top-2 maybe replaced by other experts with minimal drop in performance. Taking inspiration from this observation, the paper suggests a new method and baselines for cache-aware experts reranking. Experimental evidence is then presented to show reduced cache-misses by using the proposed cache-prior reranking algorithm.

**Audience:**

Yes

**Broader Impact Concerns:**

Nothing specific concerns noted.

**Claims And Evidence:**

No

**Requested Changes:**

- Claims that require clarification
  - Page 4, Paragraph 3, final sentence - Is Top-1 critical or Top-2 critical to maintain performance. Looking at Figure 2, it appears that Top-2 is necessary to retain (experts not be replaced) failing which perplexity gets worse for DeepSeek. Please make the argument precise.

  - Page 6, Paragraph that follows equation 8 has a discussion on uncertainty. What is meant by uncertainty here? Does the use of the term here mean that if a router outputs all logits with same value (effectively uniform distribution)  then it has highest uncertainty? The statement appears to be qualitative and somewhat confusing as uncertainty can have  multiple meanings in language models (ML models in general)

  - Page 9: the claim on improved cache consistency leading to better performance is attributed to implicit regularization. This explanation needs to be clarified as the improved score could very well be due to an entirely different (and unknown) reason

- Doe the points shown in Figures have any uncertainty information available? In other words, are these point estimates or averaged over multiple trials etc?

- There is a mention of cache-miss in text for Figure 8. I do not see any metric that can help me see cache miss in Figure 8 in the draft available for review. Please clarify what I might be missing here

- Suggestion to aid with paper completeness - please provide a description of the 4 models used in empirical analysis including training details like top_k, granularity etc (as applicable) in the appendix

**Strengths And Weaknesses:**

# Strengths

- The paper studies MoE deployments in systems with constrained resources (memory and perhaps compute)
- The observation that dropping/pruning experts with lower scores do not affect the performance in 4 popular MoE models
- The paper proposes sensible baselines in addition to a new algorithm for cache-prior reranking method

# Weaknesses

- The paper lacks a systematic study of training hyperparameter choices that can have an influence over the proposed cache-prior reranking method. The use of open source models is a good start but the reader (like this reviewer) may wonder what choices can be made during training that can assist with producing deployment-friendly models. Choices like ```top_k```, ```granularity``` would be interesting to study but understandable that its out of scope for this paper.

- Related to above, the reader may wonder whether the findings in the paper can extend to newly developed and/or released MoEs. Is there a limit to the scale of the model beyond which the cache-prior reranking might not work?

- The proposed method, cache-prior reranking, is completely heuristic. The paper provides no intuition on why this method should be better than a baseline that seemingly provides an upper bound for caching.

- The notation used to describe the baselines (Equations 5 - 7) are somewhat confusing to this reader. Could the authors consider adding more text or an example in the appendix to help the reader mechanically understand how proposed baselines and algorithms work?

- There are certain claims in the paper that needs more explanation or clarification. Please check requested changes section for more information

---

> ### Author Response · Authors · 2025-05-05
>
> We thank the reviewer for their thorough evaluation and constructive feedback. We have carefully considered all your comments and suggestions and made corresponding improvements in the revised manuscript, which we have uploaded with all changes highlighted in green for your convenience.
>
> **Lack of Systematic Study of Training Hyperparameters** We agree that understanding the impact of training hyperparameters on cache-friendliness is an important direction. However, our work focuses on deployment-time strategies and exclusively uses publicly available, pre-trained MoE models; we do not retrain or fine-tune these models. As such, systematically evaluating the effect of training-time choices (e.g., varying top-$k$ or granularity during training) would require retraining each model under different configurations, which is beyond the scope of this study.
>
> Nevertheless, we provide empirical observations on how architectural choices affect cache-friendliness at inference time. We added these insights in the revised manuscript in the new *section 4.7* and explicitly stated the limitations regarding training-time hyperparameter analysis.
>
> > - **Granularity** Our results suggest that model granularity influences cache efficiency, particularly when combined with our cache-prior reranking method. We observe that granular MoEs (Qwen-MoE, DeepSeek-MoE) tend to be more resilient to the approximate routing introduced by expert swapping compared to non-granular models like Mixtral. For instance, while Qwen-MoE and Mixtral exhibit comparable initial LRU cache miss rates (35\% and 40\%, respectively) in *Fig 4*, our cache-prior method halves Qwen-MoE's miss rate with only a 0.5\% perplexity increase. Achieving a similar miss rate reduction for Mixtral incurs a much higher perplexity penalty (2.9\%). DeepSeek-MoE demonstrates even greater resilience, halving its baseline miss rate with a negligible perplexity increase (~0.1\%).
> >- **Expansion Rate** The expansion rate, defined as the ratio of activated expert parameters to the total number of expert parameters, also appears correlated with cache performance. We observe that models with lower expansion rates generally exhibit lower baseline cache miss rates. Specifically, Phi-MoE, Qwen-MoE, and DeepSeek-MoE (all with an expansion rate of 0.125) tend to show better cache characteristics under LRU compared to Mixtral (expansion rate of 0.25).
> >- **Top-K** We observed no clear link between Top-K and overall cache-friendliness (e.g., Mixtral vs. Phi, both $k=2$). However, models with smaller $k$ (Mixtral, Phi) are more sensitive to Top-1 expert swaps and benefit more from guaranteeing Top-1 expert loading (*Fig 2* and *Fig 4*). In contrast, Qwen ($k=4$) and DeepSeek ($k=6$) show greater resilience to expert swapping, although their use of shared experts might also contribute to this behavior.
>
> **Generalization to New MoE Architectures and Scale** Our empirical study includes four diverse MoEs released at different times, representing architectures - differing in granularity, routing mechanisms, and use of shared experts - spanning multiple generations of MoE design. The consistent effectiveness of our cache-prior reranking across these architectures suggests strong generalizability to future MoE models, provided their routing and deployment paradigms remain similar. While we cannot rule out the possibility of future architectural changes that might affect cache behavior, the present diversity in our evaluation set indicates that our method is robust across current design choices.
>
> Regarding scalability, our ablation studies (*Sec 4.7, Fig 10 & 14*) show that the cache-prior reranking remains effective across a wide range of cache sizes and model scales. While extremely large models may introduce new challenges, our approach is fundamentally adaptable, as it does not require retraining or architectural changes.
>
> **Notation and Explanation of Baselines** We have added an intuitive description of the two approaches and a step-by-step example in *Appendix B*, along with a visualization (*Fig 12*) to aid understanding.
>
> **"Uncertainty" is used loosely** We agree that the use of "uncertainty" was not sufficiently precise and could be misinterpreted. We have reworded this to "confidence" to better reflect the intended meaning.  Specifically, when the router outputs are close to uniform (i.e., similar logits for all experts), the model exhibits low confidence in expert selection, as the reviewer correctly inferred.
>
> **Clarification regarding Implicit Regularization and Cache Consistency** Indeed, attributing the improved performance solely to implicit regularization is speculative. The original text stated that this is a "potentially" attributed effect. We have rephrased the sentence stating that:
>
> > "We hypothesize this may stem from an implicit regularization effect, where greater temporal consistency in expert selection stabilizes predictions, although other factors likely contribute."

---

> ### Author Response · Authors · 2025-05-05
>
> **Heuristic Nature of Cache-Prior Reranking** We appreciate the opportunity to clarify the intuition and theoretical motivation behind our method.
> - **Nature of the Heuristic** Our cache-prior reranking is indeed a heuristic, but it is grounded in empirical observations that MoE models are robust to deviations in lower-ranked expert selection (*Section 2.3, Figure 2*). This flexibility allows us to bias expert selection toward those already in cache, thereby improving cache locality.
> - **Why It Outperforms Oracle Caching:** Conventional caching methods (e.g., LRU, Belady’s optimal) are *lossless*: they do not alter the model’s routing decisions and thus cannot surpass the theoretical cache hit rate bound set by the model’s own routing. In contrast, our approach introduces *controlled, lossy* reranking-allowing the model to swap lower-ranked experts for in-cache ones, at the cost of a small, tunable increase in perplexity. This design enables us to trade a negligible amount of accuracy for a substantial gain in cache efficiency, even surpassing the oracle bound (*Figure 9*).
> - **Theoretical Justification:** By introducing a trade-off parameter ($\lambda$) in *equation 9*, our method can interpolate between the original routing (no loss, baseline cache hit rate) and a fully cache-driven selection (potentially zero cache misses, but higher perplexity). This flexibility is not available to lossless caching strategies. The key finding is that, with only a small (e.g., <1%) increase in perplexity, our cache-prior method consistently outperforms the theoretical optimal caching bound across different cache sizes and models. This demonstrates that, in practice, allowing approximate routing yields substantial system-level gains with negligible impact on model quality.
>
> We have expanded the discussion in the manuscript (*Section 3.3* and *4.8*) to clarify the rationale and practical benefits of our approach.
>
>
> **Top-1 vs. Top-2 Criticality** We agree that this distinction should be made clearer. Our results show that pruning or swapping the Top-1 expert leads to catastrophic performance degradation for all models, while swapping experts ranked 2 or lower results in a gradual performance trade-off. When considering only the top-1 expert, the right plot shows that swapping the top-1 expert with a random expert is completely ineffective for all models (Perplexity goes off the grid for all models).
>
> Regarding the Top-2 Expert, considering the DeepSeek model, we observe that pruning the top-2 experts does increase the perplexity from the original value of ~7 up to ~13 (*Figure 2 left*). In contrast, when we swap the second best expert with a random expert, we can recover much of the performance loss with a perplexity ~8.5 (*Figure 2 right*), which is still a significant performance drop.
>
> In our paper, for Mixtral and Phi-MoE models which perform Top-2 selection, we set $J=1$ and hence only allow swapping of the 2nd expert. For the granular Qwen-MoE and DeepSeek-MoE architectures which perform Top-4 and Top-6 expert selection, we set $J=2$. This was not clear from the text and was only shown in *Figure 4*. We now clearly added this to the implementation details and enhanced the presentation in *Section 2.2*.
>
> **Point estimates or averaged** All reported results are point estimates, obtained from a single pass over the entire dataset. As our algorithm is deterministic, repeated runs yield identical results. We have clarified this in the Experimental Setup *Section 4.1*.
>
> **Cache-miss is mentioned in the text for *Figure 8*, but not shown in the figure.** Thank you for catching this oversight. The figure reports only parameters loaded from Flash, and the mention of cache-miss-rate was an error. We have corrected the text.
>
> **Description of the four models used, including training details like top_k, granularity, etc.** The requested architectural details, including top-k, granularity, shared expert usage, and number of parameters, are provided in *Table 1*. If there are other architectural details that are relevant, we are happy to include them in the appendix. However, please note, as stated above, we do not retrain/train our own MoE models and some training details might not be provided by the model creators. Please let us know what other specific information or training details would be useful to include, and we would be happy to add them to the appendix.
>
> Finally, we thank the reviewer again for their thoughtful comments, which have helped us improve the clarity and rigor of our manuscript.

---

> ### Comment · Reviewer_8bhs · 2025-05-13
> **Thank you and minor clarifications**
>
> > Expansion Rate The expansion rate, defined as the ratio of activated expert parameters to the total number of expert parameters, also appears correlated with cache performance. We observe that models with lower expansion rates generally exhibit lower baseline cache miss rates. Specifically, Phi-MoE, Qwen-MoE, and DeepSeek-MoE (all with an expansion rate of 0.125) tend to show better cache characteristics under LRU compared to Mixtral (expansion rate of 0.25).
>
> Based on above, expansion rate appears to be another name for sparsity which is the commonly used term in MoEs. Interesting to note that sparser models tend to be more cache friendly.
>
> > Why It Outperforms Oracle Caching: Conventional caching methods (e.g., LRU, Belady’s optimal) are lossless: they do not alter the model’s routing decisions and thus cannot surpass the theoretical cache hit rate bound set by the model’s own routing. In contrast, our approach introduces controlled, lossy reranking-allowing the model to swap lower-ranked experts for in-cache ones, at the cost of a small, tunable increase in perplexity. This design enables us to trade a negligible amount of accuracy for a substantial gain in cache efficiency, even surpassing the oracle bound (Figure 9).
>
> I believe the above statement on tradeoff should be qualified by he fact that the model performance is tested only on GSM8K. The authors may want to avoid making strong statements on tradeoff between cache efficient and accuracy.
>
> Overall, I thank the authors for their rebuttal that has addressed most of my review questions and comments.

---

> > ### Author Response · Authors · 2025-05-13
> >
> > We sincerely thank the reviewer for their positive feedback and insightful suggestions, and for acknowledging that most of their previous comments have been addressed.
> >
> > Regarding the Trade-off Statement (Cache Efficiency vs. Accuracy):
> > We would appreciate clarification on the reviewer’s suggestion to avoid strong statements about the trade-off between cache efficiency and accuracy.
> >
> > To clarify, this trade-off has been evaluated across all three benchmarks: WikiText (language modeling), MMLU, and GSM8K, with Pareto fronts presented in Figures 4, 5, and 6. These results demonstrate that our Cache-Prior method enables a controlled trade-off between accuracy and cache efficiency by adjusting the $\lambda$ parameter. This tunable trade-off is a central design principle of our approach.

---

> > > ### Comment · Reviewer_8bhs · 2025-05-13
> > >
> > > > Regarding the Trade-off Statement (Cache Efficiency vs. Accuracy): We would appreciate clarification on the reviewer’s suggestion to avoid strong statements about the trade-off between cache efficiency and accuracy.
> > >
> > > > To clarify, this trade-off has been evaluated across all three benchmarks: WikiText (language modeling), MMLU, and GSM8K, with Pareto fronts presented in Figures 4, 5, and 6. These results demonstrate that our Cache-Prior method enables a controlled trade-off between accuracy and cache efficiency by adjusting the
> > >  parameter. This tunable trade-off is a central design principle of our approach.
> > >
> > > I thank the authors for seeking this clarification. My thought was that reminding the reader that the paper uses few-shot downstream evaluations (as opposed to RL etc) to make an argument about cache efficiency vs accuracy tradeoff would be helpful (to the reader). If the authors think this is unnecessary I respect that and will not use this point to argue against the paper.
> > >
> > > Overall, the paper is a good exploration of how a simple (heuristic) method can help with MoE use on devices with batch size = 1.

---

### Review · Reviewer_SKhL · 2025-04-27

**Summary Of Contributions:**

MoE models have recently become highly powerful, but their growing parameter counts make deployment under memory-constrained settings challenging. Cache-based methods have emerged as a mitigation strategy. In this paper, the authors observe that modifying an expert selection when the change does not involve the highest-probability expert has a smaller impact on model performance compared to pruning experts outright. Based on this insight, they propose using a cache prior to adjust expert weights, encouraging selection of experts already present in the cache. Experimental results demonstrate that this approach can significantly reduce the cache miss rate with only a small degradation in perplexity or accuracy.

**Audience:**

Yes

**Claims And Evidence:**

Yes

**Requested Changes:**

- The placement of Figure 1 can be optimized. Currently, it appears on Page 1, while the corresponding explanation for the left part of the figure only appears on Page 3 (left)/Page 10 (right). Additionally, the different colors and arrows used in the left panel of Figure 1 are not explained at all. The authors should improve the figure's placement and make Figure 1 more self-explanatory, including a clear legend or caption describing the color scheme.

- The description of Figure 2 is unclear. The figure caption should explicitly state that the dashed line represents the baseline.

- Citation format needs to be carefully checked. E.g. Page 2 DeepSeek V3, OLMoE, etc. Please change the \cite command into \citep.

- The authors state: "A key issue is that state-of-the-art MoEs lack temporal locality in expert selection, leading to inefficient LRU caching due to frequent evictions and reloads, which results in a poor cache hit rate."
However, "temporal locality" is not formally defined in the paper. Moreover, several existing works have noted similarities in expert assignment across sequential decoding steps. The authors should either provide experimental evidence to support this claim or cite appropriate references.

- Please refer to the Weaknesses section of this review for additional specific comments regarding figure readability, equation clarity, and text simplification.

- *The authors are encouraged to open-source the implementation of the proposed algorithm.*

**Strengths And Weaknesses:**

Strengths:

- The proposed algorithm is simple and intuitive.
- Extensive experiments are conducted across a wide range of state-of-the-art MoE-based LLM architectures and tasks. The results comprehensively show the effectiveness and robustness of the method in reducing cache miss rates.

Weakness:
- The writing quality needs significant improvement. Many sentences are unclear, and the overall flow could be much more polished.

- Formatting issues, such as improper citation styles, suggest a lack of careful proofreading, raising concerns about the overall attention to detail.

- Equation 5 lacks clarity. Since sets are inherently unordered, the operation described appears to be a no-op. The authors should consider representing the experts as a list or another ordered data structure to make the operation meaningful.

- Section 3.2, as I understand correctly, is just the top-p sampling from [1], a widely known and commonly used method in LLM-related work. This section could be greatly simplified or merged into the background.

- The text in Figure 3 is too small and difficult to read. Furthermore, the meaning of the different colors is not clearly explained.

- Figures 5 and 6 lack sufficient explanation. In particular, it is unclear why some data points are not connected. More context and description should be provided to help the reader interpret the results.

[1] Holtzman, Ari, et al. "The curious case of neural text degeneration." ICLR 2020

---

> ### Author Response · Authors · 2025-05-05
>
> We thank the reviewer for the detailed and constructive feedback. We appreciate the remarks regarding the simplicity of our algorithm and the comprehensive experiments demonstrating its effectiveness across various MoE-based LLM architectures.
>
> We have carefully considered all your comments and suggestions and made corresponding improvements in the revised manuscript, which we have uploaded with all changes highlighted in purple for your convenience.
>
> **Response to General Comments on Writing and Citation** We thank you for pointing out the issues regarding writing quality, clarity, and citation formatting. We have thoroughly proofread the entire manuscript to improve sentence clarity and overall flow. We also fixed all citation formatting issues, replacing \cite commands with \citep as appropriate.
>
> **Equation 5 Clarity** We agree that the original presentation of Equation 5 could cause confusion because sets are typically unordered. In the revised manuscript, we clarified that the sets used here are *ordered sets* that maintain the order of elements as in their original ranking. This ensures the promotion operation is well-defined. We added the following clarifying sentence:
> > *"Importantly, all sets in this context are ordered sets and maintain the order of elements as in their original ranking. This ensures that the promotion operation is well-defined, as it preserves the relative ordering of experts throughout."*
>
> Additionally, we included a more intuitive explanation of the baselines along with an example and schematic visualization in *Appendix B*.
>
> **Section 3.2 and Relation to Top-p Sampling** We appreciate the observation regarding the similarity to Top-p sampling from Holtzman et al. (ICLR 2020). However, we believe that simply merging or simplifying Section 3.2 to refer to Top-p sampling would risk confusing readers. Our use case differs because Top-p sampling limits token probabilities for generation, whereas our method limits the number of experts considered for potential swapping, followed by a max-rank approach. While the concept of deriving a dynamic candidate set is similar, the subsequent steps and objectives differ substantially. To address this, we have included a brief note in the revised manuscript acknowledging the conceptual similarity in obtaining the dynamic set.
> > We address this issue in the cumulative probability threshold approach by dynamically choosing the max-rank
> M for every layer and every input x. The process is conceptually similar to the sampling approach by
> Holtzman et al. (2020) in which tokens are sampled from a dynamic set containing the vast majority of the
> probability mass. We determine M by summing...
>
> **Figures** Thank you for the feedback. We addressed all comments and incorporated the suggestions:
>
> - *Figure 1*: We improved the placement of Figure 1 so that the explanation aligns better with the figure’s appearance. We also added a clear legend and caption describing the color scheme to make it more self-explanatory.
>
> - *Figure 2*: The caption now explicitly states that the dashed line represents the baseline.
>
> - *Figure 3*: We increased the font size of the text to enhance readability and added a detailed explanation of the meaning of the different colors used.
>
> - *Figures 5 and 6*: We expanded the figure captions and the main text to provide more context and explanation, including clarifying the Pareto front curve and why some data points are not connected.
>
> **Temporal Locality Definition and Evidence**  We have added a formal definition of *temporal locality* in the Introduction to clarify the concept. Furthermore, we pointed to *Section 4.6* in the manuscript, where we provide empirical evidence supporting our claim. Specifically, Table 2 reports the *average lifetime* of experts in cache (the average number of time steps an expert remains in memory), which clearly demonstrates the lack of temporal locality in state-of-the-art MoEs and motivates our cache-aware routing approach.
> > Temporal locality, in the context of MoE caching, refers to the principle that recently accessed experts are likely to be accessed again in subsequent token generations. A key challenge, which we analyze empirically in Section 4.6 is that state-of-the-art MoEs often lack strong temporal locality in expert selection, leading to inefficient Least Recently Used (LRU) caching strategies due to frequent evictions and reloads, which results in a poor cache hit rate.
>
> **Open-Sourcing Implementation** We appreciate the encouragement to open-source our implementation. We are currently considering to release the code publicly.
>
> We thank the reviewer again for their thoughtful comments, which have helped us improve the clarity and rigor of our manuscript.

---

> ### Comment · Reviewer_SKhL · 2025-05-12
>
> I thank the authors for improving the overall clarity and readability of this paper. I think most of my concerns have been addressed properly. Here are some further suggestions:
>
> Figure 1: Please add legend for the router and cache prior to show that these components are also stored in DRAM.
>
> Temporal Locality Definition and Evidence: For Table 2, it will be great if the author can also report the standard deviation of the cache lifetime.

---

> > ### Author Response · Authors · 2025-05-13
> >
> > We sincerely thank the reviewer for their positive feedback and insightful suggestions. We are pleased that most of their concerns have been addressed.
> >
> > - Regarding *Figure 1*, we have updated the legend in Figure 1 to explicitly indicate that static model parameters, including the router weights, reside in DRAM. Additionally, we have changed the color of the router and Cache-Prior blocks to blue for better consistency with this labeling.
> >
> > - Thanks for this suggestion. We agree and we are currently running these experiments and will update Table 2 with the standard deviation values accordingly in the revised manuscript.

---

### Decision · Action_Editor_b5ZF · 2025-05-23

**Recommendation:** Accept as is

**Comment:**

This paper presents an algorithm that aims to make routing more consistent by exploiting the fact that MoEs can remain effective even if the "default" top-k routing decisions are perturbed. By making routing more consistent, MoEs can be made more applicable and efficient in resource-constrained settings, since repeated loads/swapping of active experts can be avoided. The paper evaluates on a few standard LLM benchmarks and presents compelling efficiency gains with minimal performance degradation. Reviewers all were unanimous in recommending acceptance. One reviewer suggested considering open-sourcing the algorithm to support future research.

**Audience:**

Yes, making MoE models more applicable in resource-constrained scenarios is almost certainly of interest to members of the TMLR community.

**Claims And Evidence:**

All reviewers felt that the main claim of this work (the presentation of a cache-aware routing strategy that aims to reuse experts) was supported by the evidence (a series of experiments on a few standard benchmarks that measure both performance and resource efficiency).